

# Source apportionment of the submicron organic aerosols over the Atlantic Ocean from 53 N to 53 S using HR-ToF-AMS

Shan Huang[1,2,†], Zhijun Wu[3,†], Laurent Poulain[2], Manuela van Pinxteren[2], Maik Merkel[2], Denise Assmann[2], Hartmut Herrmann[2], Alfred Wiedensohler[2]

[1]Institute for Environmental and Climate Research, Jinan University, Guangzhou, 511443, China
[2]Leibniz Institute for Tropospheric Research, Leipzig, 04318, Germany
[3]College of Environmental Sciences and Engineering, Peking University, Beijing, 100871, China
†: Shan Huang and Zhijun Wu contribute equally.

*Correspondence to*: Laurent Poulain (poulain@tropos.de), Shan Huang (shanhuang_eci@jnu.edu.cn)

**Abstract.** The marine aerosol is one of the most important natural aerosol systems and can significantly impact the global climate as well as biological cycle. A series of measurements during four open-ocean cruises in 2011 and 2012 over the Atlantic from 53 N to 53 S were conducted to reveal the physical and chemical properties of the marine boundary layer (MBL) aerosol and its seasonality. Chemical composition of the submicron particles was obtained using the on-line techniques High Resolution Time-of-Flight Aerosol Mass Spectrometer (HR-ToF-AMS) as well as from offline high-volume PM$_1$ filter samples with a sampling time of 24 hours. Our measurements show that the MBL aerosol particle mass is controlled by sulfate (50%), followed by organics (21%), sea salt (12%), ammonium (9%), Black Carbon (BC, 5%) and nitrate (3%). Only sulfate exhibits pronouncedly seasonal dependency, while no such trend was observed in other species. Source apportionment of the organic fraction was performed using Positive Matrix Factorization (PMF). Five factors were identified, including three marine sources and two non-marine sources. Marine sources are linked to primary production (19% of total organic aerosol (OA) mass), marine dimethylsulfide (DMS)-oxidation (16%), and amine-related secondary formation (16%). The other two OA components are attributed to continental outflow (19%) and aged ship exhausts - biomass burning emissions (30%). Our study indicates that, on average, non-marine sources nearly have the equal importance to the Atlantic aerosols comparing with the marine sources, respectively contributing 49% and 51% to the total OA mass loadings. The South Atlantic atmosphere is found to be less polluted than the North according to our source analysis. Detailed latitudinal distribution of OA sources showed that DMS oxidation contributes remarkably to the MBL aerosol over the South Atlantic during spring, while continental pollutants largely contaminate the marine atmosphere when near the west and middle Africa (15 N~15 S) as well as Europe. Based on our measurements, SOA produced from DMS oxidation over the Atlantic can be estimated as MSA mass concentration times a scaling factor 1.79 for spring season, which is derived from the strong correlation (R$^2$ >0.85) between MSA and DMS-oxidation OA component.



# 1 Introduction

As one of the most important natural aerosol systems at the global level, the marine aerosol plays a significant role in the global radiation budget through both direct and indirect climate effect, as well as in biogeochemical cycling (O'Dowd and De Leeuw, 2007; Saltzman, 2009). The marine aerosol in pristine conditions, includes the primary products, *i.e.* sea spray aerosol (SSA), mechanically generated via sea/air interaction (Andreas, 2002) and secondary particles, which are chemically produced from atmospheric reactions of gases emitted from the organisms in the ocean (Charlson et al., 1987). The nucleation events were observed in the boundary layer in coastal measurements, e.g. Mace Head (O'Dowd et al., 2010; Rinaldi et al., 2009), and the Arctic Ocean (Leck and Bigg, 2010). However, over the open "warm" ocean, the new particle formation in the marine boundary layer (MBL) was not found. It is believed to take place in the free troposphere, in which the resulting new particles (with diameter 3 to 10 nm) grow to Aitken mode (10 – 100 nm) size range by condensation, and then are entrained to the MBL (Heintzenberg et al., 2004). For a long time, the focuses of marine aerosol research have been put on the sea salt (de Leeuw et al., 2011; de Leeuw et al., 2000; Geever et al., 2005; Grythe et al., 2014; Lewis and Schwartz, 2004; Ovadnevaite et al., 2012; Sofiev et al., 2011; White, 2008), and on sulfate (Ayers and Gillett, 2000; Bates et al., 1987; Charlson et al., 1987; Gondwe et al., 2003; Ole Hertel 1994; Shiro Hatakeyama, 1985; von Glasow and Crutzen, 2004). Recently, ambient measurements found that organics could be a dominating component of submicron marine aerosol particles, contributing up to 77% of the total particle mass concentration during phytoplankton bloom periods (O'Dowd et al., 2004; Ovadnevaite et al., 2011). Also, Quinn and Bates (2011) pointed out that organic aerosols are a considerable source of the cloud condensation nuclei (CCN), challenging the hypothesis that MBL cloud process is only controlled by the sulfate from dimethylsulfide (DMS)(Charlson et al., 1987). However, the chemical and physical characteristics of the organic component in the marine aerosol are still less understood.

The marine organic aerosols have been found to generally contain carboxylic acids, organic hydroxyl groups, alkane groups, as well as organosulfate (Claeys et al., 2010; Decesari et al., 2011; Hawkins et al., 2010; Russell et al., 2010; Schmitt-Kopplin et al., 2012). Carboxylic acids could compose nearly 30% of the total submicron organic mass over the ocean, as important as alkane group and organic hydroxyl (Hawkins et al., 2010; Russell et al., 2010). Yet, there are still a large fraction of the oxygenated compounds remaining unidentified (Decesari et al., 2011). Methanesulfonic acid (MSA) is one of the few compounds that can be identified and quantified. MSA is almost exclusively formed from the oxidation of DMS that is emitted from phytoplankton in the ocean (Becagli et al., 2013; Gondwe et al., 2004). This makes MSA a good indicator for secondary organic aerosols (SOA) with marine origin. Besides DMS, several biogenic volatile organic compounds such as isoprene and monoterpenes have been recognized as possible precursors for marine SOA (Fu et al., 2011; Hu et al., 2013), though their significance in SOA formation remains unclear because inconsistent conclusions were drawn in different studies (Arnold et al., 2009; Meskhidze and Nenes, 2006). It is, however, difficult to distinguish the primary and secondary marine organic aerosols based on chemical characteristics. For example, carboxylic acids could be the highly oxygenated products via secondary pathway, but also have been recognized as a distinct component in the marine primary organic matters




(Hawkins and Russell, 2010); amines can contribute up to 14% of the submicron OA mass in the fresh sea spray aerosols (Quinn et al., 2014; Russell et al., 2010), and can also serve as precursor of marine SOA (Facchini et al., 2008; Ge et al., 2011; Müller et al., 2009).

Although certainly contributed by the ocean, the MBL aerosols can be influenced by many non-marine sources, including continental pollutants and ship emissions (Andreae, 2007; Heintzenberg et al., 2000; Saltzman, 2009; Simpson et al., 2014). Frossard et al. (2014) observed that organic matters were impacted by ship and continental pollutants during 63% of the sampling time in a series of measurements over the Pacific and Atlantic. Over the central Arctic Ocean, the identified continental contribution is comparable to the marine biogenic sources (36% vs 33%), and the rest 31% of the sampled ambient aerosol mass can possibly be influenced by multiple sources including aged continental emissions (Chang et al., 2011). Also, the measurements on ship transects between the South Asia and Northern Japan indicate heavy contaminations from non-marine sources in MBL aerosols, and the OA is mainly contributed by biomass burning (BB) emissions and anthropogenic pollution (Choi et al., 2017). Nevertheless, there are still some regions with little anthropogenic impact on the marine aerosol. For example, Ceburnis et al.(2011) found 80% organic aerosol matter of biogenic origin related to marine plankton emissions over the northeast Atlantic. O'Dowd et al.(2014) pointed out that the anthropogenic and coastal effects can be sufficiently minimized with the selection criteria to ensure the predominant marine contribution to the measured aerosols at Mace Head, a research station in the coast of the northeast Atlantic.

So far, the source apportionment of the MBL aerosols has been performed with multiple indicators or methods such as isotopes (Ceburnis et al., 2011; Seguin et al., 2011; Seguin et al., 2010), transmission electron microscope (e.g. Bigg and Leck, 2008; Leck and Bigg, 1999; Leck and Bigg, 2005), cluster analysis (Frossard et al., 2014; Rebotier and Prather, 2007), principal component analysis (Decesari et al., 2011), and positive matrix factorization (PMF) method (Chang et al., 2011; Choi et al., 2017; Schmale et al., 2013a). But, many details of the source contribution to the MBL aerosol are still unknown at both temporal and spatial scales. This is because the observations over the open ocean are few (Choi et al., 2017; Dall'Osto et al., 2010; Zorn et al., 2008) and most of the marine aerosol measurements were conducted on coastal or island stations (e.g. Crippa et al., 2013b; Ovadnevaite et al., 2014; Rinaldi et al., 2010; Schmale et al., 2013a).

In this study, aerosol measurements were conducted on board German research vessel (*R/V*) Polarstern based on 4 cruises over the Atlantic Ocean in 2011 and 2012, covering the range from 53 °N to 53 °S. The detailed chemical characteristics of the Atlantic aerosols were measured by a high resolution time-of-flight aerosol mass spectrometer (HR-ToF-AMS). Based on this unique data-set, sources of the organic aerosol particles over the major part of the Atlantic were investigated in a high temporal and spatial resolution to provide a latitudinal distribution of OA source contributions in spring and autumn.



## 2 Methods

### 2.1 Ship-board campaigns measuring submicrometer marine aerosols over the Atlantic

The aerosol measurements were conducted by Leibniz Institute for Tropospheric Research (TROPOS) during four cruises on board of RV Polarstern in 2011 and 2012. Expedition details and ship tracks of the cruises are shown in Table 1 and Figure

1. All cruises were part of transfer voyages between Bremerhaven, Germany (53°33′N, 8°35′E) and the German Antarctic research station, via either Cape Town, South Africa (33 °55′S, 18°25′E) or Punta Arenas, Chile (53°10′S, 70°56′W). In each cruise, the ship shortly moored at Las Palmas, Spain (28°9′N, 15°25′W) for a supply. The cruises can be divided into two groups according to the campaign period of the year: Cruise 1 (CR1) and Cruise 3 (CR3) correspond to spring in the Northern Hemisphere (NH), while Cruise 2 (CR2) and Cruise 4 (CR4) correspond to autumn in the NH. In terms of the

spatial range, CR1, CR2 and CR4 followed almost the same route, while CR3 had the different track in South Hemisphere, but followed the same route than the other three cruises from ~ 15 °N to 53 °N (Figure 1).

During all four cruises, the instruments were deployed inside (and on the roof) of an air-conditioned container, which was located on the first deck of the vessel, approximately 30 m above the ocean surface (Figure 1). The whole-air sampling inlet of the container was made of a stainless-steel tube (6 m long, 40 mm diameter, with an inclination angle of 45 ° to the

15 container roof). A vacuum system kept a stable total aerosol flow rate of approximately 15 liter per minute (l min$^{-1}$) through the inlet. The inlet loss due to diffusion and deposition was calculated by the tool Particle Loss Calculator (von der Weiden et al., 2009). According to this tool, the container inlet completely excluded particles with mobility diameter larger than about 3 μm. In submicrometer size range (10 nm - 1000 nm), the average inlet efficiency is nearly 100%, but decreases rapidly for particles smaller than 2 nm and larger than 1500 nm (Figure S1).

An isokinetic splitter was used downstream of the aerosol inlet to distribute the aerosol flow to different online instruments for aerosol physicochemical properties as mentioned below. Particle size distributions from 10 nm to 3 μm (mobility diameter, equal to volume equivalent diameter for spherical particles) were provided by a combination of one TROPOS-type mobility particle size spectrometer (MPSS, 10 nm - 800 nm, see e.g. Wiedensohler et al., 2012), and one Aerodynamic Particle Size Spectrometer (type TSI APS model 3321, 500 nm - 10 μm, TSI Inc., USA). The particle hygroscopicity at 90 %

relative humidity (RH) was measured by a Hygroscopicity Tandem Differential Mobility Analyser (HTDMA, see details in e.g. Massling et al., 2011; Wu et al., 2011). A Differential Mobility Analyser – Cloud Condensation Nuclei Counter (DMA-CCNc, see e.g. Henning et al., 2014) was used for size-resolved measurements of cloud condensation nuclei. Optical properties including light scattering and absorption coefficients of particles were provided by an Integrating Nephlometer (Model 3563, TSI Inc., USA), a Multi Angle Absorption Photometer (MAAP, Model 5012, Thermo Inc. USA) and a Particle

Soot Absorption Photometer (PSAP, Radiance Research, Inc., Seattle, WA, USA). Particle chemical properties were investigated using a HR-ToF-AMS (Aerodyne Research, Inc., USA), which is the central instrument for the present study. In parallel, an offline particulate matter (diameter < 1 μm, i.e., PM$_1$) sampler (Digitel, DHA-80, Digitel Elektronik AG,



Switzerland) that was fixed on the roof of the aerosol container provided daily (24 h) particle chemical composition. Before each campaign all instruments in the aerosol container were synchronized to UTC time (Coordinated Universal Time).

## 2.2 Measurements

### 2.2.1 Particle chemical analysis

#### 2.2.1.1 HR-ToF-AMS

HR-ToF-AMS (referred to as AMS in the following text) can directly distinguish the elemental composition of ions having the same nominal mass (DeCarlo et al., 2006). The instrument principle and field deployment have been described in detail in previous publications (e.g. Canagaratna et al., 2007; DeCarlo et al., 2006; Drewnick et al., 2005; Jayne, 2000; Zhang et al., 2007). The particles drawn into AMS impact onto a thermal vaporizer plate (600 ℃) where the non-refractory (NR) part of

the particles were detected. The AMS provides size resolved chemical composition of aerosol particles in submicrometer size range(Canagaratna et al., 2007). Its size cut is within the range of nearly unity transmission efficiency of the main container inlet during Polarstern cruises (Figure S1). The inlet flow rate of AMS was approximately 0.08 l min⁻¹. To minimize the inlet loss between the AMS and MPSS, the AMS was located next to the MPSS and the inlet of AMS was connected direct in front of the MPSS inlet. The two instruments shared a Nafion dryer to maintain relative humidity (RH)

lower than 40 %. The AMS was operated alternatively between V- and W-modes associated with PToF and/or MS modes at a time resolution of 2 min. Collection efficiency (CE) of 0.7 was determined based on inter-comparisons between: (1) AMS and MPSS, (2) AMS and offline measurements (Huang et al., 2017).

The default components measured by AMS include organics, sulfate, nitrate, ammonium and chloride. Particularly, marine biogenic tracer MSA was quantified with the AMS based on standard calibrations and validated by collocated offline

measurements (Huang et al., 2017). The relative ionization efficiencies (RIE) of the above species were calibrated by the standard tests using pure ammonium nitrate, ammonium sulfate and MSA weekly during the measurements. In order to better reduce the signal noise, AMS data in 20-min average are calculated and used in the following analysis except as noted. Table 2 provides the detection limits (DL) of detected AMS species for 20-min resolution (calculation details see Huang et al., 2017). Besides, a total uncertainty of ~ 30 % is estimated for AMS measurements, including 10 % for the inlet system,

20% for the ionization efficiency calibration and 20 % for the collection efficiency (Crippa et al., 2013a; Freutel et al., 2013; Poulain et al., 2014).

The AMS data during Polarstern measurements were analyzed using the software Squirrel v1.54 for the unit resolution and Pika v1.13 for the high-resolution, both downloaded from the Tof-AMS webpage (http://cires1.colorado.edu/jimenez-group/wiki/index.php?title=ToF-AMS_Main ). The software was based in Igor Pro (WaveMetrics Inc., version 6.22A).

#### 2.2.1.2  Offline measurements

A PM₁ high volume Digitel filter sampler was deployed to sample aerosol particles at 24-hour time resolution (midnight to midnight, UTC) during Polarstern CR1, CR2 and CR3, working at flow rate of 500 l min⁻¹. The daily aerosol masses were



collected on quartz fiber filters (150 mm, Munktell, MK 360, Bärenstein, Germany), which were pre-treated at 105 ℃ for 24 h before the measurement. All filter samples were stored in the fridge at -20 ℃ until being analysed. The total aerosol particle mass was determined by the weight of the clean and particle loaded filter, and the loaded filter was separated in several aliquots for different analysis. Inorganic ions and oxalate were measured after aqueous filter extraction (25% of the filter in 20 mL, filtered with a 0.45 μm syringe) with ion chromatography (IC, ICS3000, Dionex, Sunnyvale, CA, USA). Organic carbon (OC) and elemental carbon (EC) were analyzed by a thermographic method (C-mat 5500, Ströhlein, Germany) using EUSAAR 2 protocol (see more details in van Pinxteren et al., 2017). In total 86 ambient samples from CR1 (25 samples), CR2 (30 samples) and CR3 (31 samples) were analysed, while 45 effective samples were used in this study due to exclusion of filters contaminated by ship exhausts and sea-water contact under stormy conditions.

### 2.2.2 Other measurements and data sources

During the Polarstern measurements, the particle number size distribution in the range between 10nm and 800nm was measured by a TROPOS-type MPSS, described in detail in previous literatures (e.g. Birmili et al., 1999; Birmili et al., 1997; Wiedensohler et al., 2012). In this study, only MPSS was used as external instrument to evaluate data quality of the AMS due to the similar measuring range of particle size (smaller than 1μm). This MPSS was operated at time resolution of 8 min (CR1, CR2) and 5 min (CR3, CR4) with condition of RH < 40 %. The resulting particle number size distribution was corrected for internal particle losses. Generally, an uncertainty of approximately 10 % can be considered as shown by inter-comparison experiments (Wiedensohler et al., 2012).

The particle mass concentration of black carbon (BC) was converted from particle light absorption coefficient provided by a MAAP. Detailed instrument description is available in previous articles (Müller et al., 2011; Petzold et al., 2002; Petzold and Schönlinner, 2004). The instrument measures the aerosol particle light absorption coefficient at 637 nm and reports the BC mass concentration by applying BC mass absorption coefficient 6.6 $m^2\,g^{-1}$. To be combined with the particle concentration measured by AMS, BC mass concentration in submicrometer size range is required. Poulain et al. (2011b) reported that BC mass concentration in $PM_1$ is approximately 90% of that in $PM_{10}$ according to the comparison between the data from 2 MAAPs with $PM_1$ and $PM_{10}$ inlet in the central European-background station of Melpitz (Germany). However, here the MAAP was connected to the main inlet of the aerosol container which had significant particle loss when particle diameter was larger than 1500nm. Thus, we suppose that the measured BC particles are almost in the submicrometer size range. In total, a global uncertainty of 5% was counted for MAAP measurements taking the uncertainties of instrument, size cutting, density and mass absorption efficiency into consideration.

All meteorological parameters on Polarstern cruises were measured by an on-board German Weather Service (Deutscher Wetterdienst, DWD) station. Air mass back trajectories along the ship track in 12-hour time resolution (00:00 and 12:00 UTC everyday) were also retrieved by the DWD, using a global meteorological model GME (Global Model of the Earth, Majewski et al., 2002). Air masses at 950 hPa (approximately 500 m) were selected and considered as a well-mixing situation; and backward trajectories in last five days (120 h) were investigated in this study. Air mass back trajectory data



were directly obtained from the DWD. Navigation parameters together with meteorological parameters were supplied by the Polarstern central data acquisition system (https://dship.awi.de/).

## 2.3 Positive matrix factorization (PMF)

PMF is frequently used for organic aerosol (OA) mass spectra analysis from AMS data, dividing OA into several factors with different origins or oxidation state (Lanz et al., 2007; Ng et al., 2011; Ulbrich et al., 2009; Zhang et al., 2011). The principle of this statistical model is described in the Equation 1 (Paatero, 1997; Paatero and Tapper, 1994):

$$x_{ij} = \sum_p g_{ip} f_{pj} + e_{ij} \tag{1}$$

In this study, the $x_{ij}$ is the element of the matrix of the measured AMS organic data to be fit (whose columns are time series of each $m/z$ and rows are mass spectra with signal intensity at $m/z$ from 12 to 120). The $g_{ip}$ and $f_{pj}$ are the elements of the matrices presenting the variability (time series) and the profile (mass spectra) for $p$ factors according to the model solution, and values in these two matrices are constrained to be positive. The $e_{ij}$ is the element of the matrix of the residuals. The subscripts $i$, $j$ and $p$ correspond to the row or column indices in the matrices.

To fit the data, the model uses a weighted least-squares algorithm based on the known standard deviations ($\sigma_{ij}$) of the elements of the data matrix to minimize a parameter $Q$, as well as minimizing the residuals (Equation 2) (Paatero, 1997; Ulbrich et al., 2009).

$$Q = \sum_{i=1}^m \sum_{j=1}^n (e_{ij}/\sigma_{ij})^2 \tag{2}$$

The advantages of PMF method are: (1) the measurement errors are considered in statistical calculation, and (2) the solutions of PMF are physically meaningful for ambient aerosol since the matrix elements are constrained to be positive.

For Polarstern AMS data, PMF analysis was run for a 4-cruise combined HR organic dataset using the PMF2 algorithm in the Igor Pro-based PMF Evaluation Tool (PET, v2.06, Ulbrich et al., 2009). Before running the PMF model, HR organic mass matrix and its error matrix were examined to remove ship contamination periods, incorrect signals (e.g. organic fragments influenced by strong sulfate signal) and very low signals (e.g. $C_x$ group, isotopes and $m/z > 120$). According to the instructions from Ulbrich et al. (2009), a minimum error estimate of one measured ion was calculated; the "bad" data with Signal-to-Noise Ratio (SNR) less than 0.2 were removed, and the "weak" data with SNR between 0.2 and 2 were down-weighted by a factor of 2; signals for $m/z$ 44 ($CO_2^+$) and related ions ($O^+$, $HO^+$, $H_2O^+$ and $CO^+$) were also down-weighted by a factor of 2 in order to avoid overestimating the importance of $CO_2^+$. The elemental analysis of PMF factors was performed by the tool Analytical Procedure for Elemental Separation (APES light v1.06) detailed by Aiken et al. (2008) and Canagaratna et al. (2015). To find the best solution, PMF was run between 1 and 7 factors for (1) rotational forcing parameter fpeak between -1 and 1 (step of 0.2), and (2) seeds with random starts between 0 and 50 (step of 2). The PMF analysis is detailed below in section 3.2.



# 3 Results and discussions

## 3.1 Particle chemical composition

### 3.1.1 AMS data quality assurance

During all four cruises, the ship's own exhausts from the main chimney affected the measurements, especially when the
wind came from behind the ship. In this study, the detailed contaminated periods are identified based on extremely high
concentration of combustion tracers (e.g. BC) and meteorological parameters (e.g. wind direction) as illustrated in Figure S2
and Figure S3. Accordingly, a ship contamination range of relative wind direction (RWD) was determined as between 135 °
and 250 °. With this criterion, 85.2% of ambient AMS data (2-min resolution) remained as "clean" data during all 4 cruises.
In the following text, all the data analysis is performed on the "clean" data except as noted.

Prior to investigating the particle chemical composition during the Polarstern measurements, the inter-comparisons between
AMS and parallel measurements were performed to assure the AMS data quality (Figure S4 and Figure S5). The total
particle mass concentration from AMS is calculated as the sum of the mass concentrations of default AMS species (i.e.
organics, sulfate, nitrate, ammonium) completed by estimated sea-salt and BC. It is compared to the calculated particle mass
concentration derived from the particle number size distribution. Taking sea salt into the consideration can slightly improve

the correlation between two different techniques (slope varies from 0.81 to 0.85; $R^2$ varies from 0.72 to 0.77, Figure S4).
Also, the mass concentration of the individual species from AMS is compared to that from the offline measurements (Figure
S5). Organic matter (OM) provided by AMS measurement is correlated with organic carbon (OC) from filter measurements
($R^2 = 0.53$). The slope of 1.75, i.e. OM/OC ratio, is similar to those found in the coastal cities as well as offshore areas, e.g.
1.7 - 1.9 in Pasadena (Hayes et al., 2013), 1.8 in the Gulf of Mexico (Russell et al., 2009), and 1.66 – 1.72 in Paris (Crippa et

al., 2013b). The excellent linear correlation is found for sulfate (slope = 1.16, $R^2 = 0.94$), and the correlation for ammonium
is moderate (slope = 0.80, $R^2 = 0.63$). No correlation is found for nitrate and chloride. A part of nitrate concentrations from
the AMS are higher than those from offline measurements, probably attributed to the evaporative losses of nitrate especially
under low mass concentration conditions and the unidentified contribution of the organic nitrates. Chloride is underestimated
by the AMS because this instrument partly detects the sea salt, which is the main source for the chloride measured during

Polarstern cruises. Considering that nitrate and chloride account for only a tiny fraction of the measured particle mass
concentration and the sea salt is estimated specifically, the comparisons above ensure the data quality of AMS measurements.
The inter-comparisons also demonstrated that a constant Collection Efficiency (CE) of 0.7 for AMS measurements could
achieve better agreement between AMS and external measurements, whereas the Composition Dependent Collection
Efficiency (CDCE) introduced by Middlebrook et al. (2012) does not seem appropriate for Polarstern measurements because

of the high acidity and low ammonium nitrate content of the MBL aerosol particles. A detailed explanation has been given in
the previous paper (Huang et al., 2017).

The important primary marine product, sea salt, was estimated using the method from Ovadnevaite et al. (2012) who
successfully quantified the sea salt aerosol concentration with the sea salt typical ion NaCl+ (m/z 57.95) and applying a

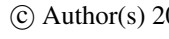



scaling factor of 51. Although NaCl$^+$ ion could be well identified by the high resolution mass spectrometer of the AMS, the correlation between NaCl$^+$ ion and sea salt derived from PM$_1$ filter samples in Polarstern measurements showed a mild relationship with R$^2$ of 0.38 (Figure S6). The possible reasons for this might be, first, the slightly different sampling (inlet) location of the two samples; and second, the small amount of the available offline data during Polarstern cruises. The

effective samples (45 samples) may be insufficient to provide meaningful correlation between offline and AMS sea salt in 4 Polarstern cruises, especially when comparing to the full year measurements in the reference. Therefore to be consistent with the literatures (Ovadnevaite et al., 2012; Ovadnevaite et al., 2014; Schmale et al., 2013a), the scaling factor of 51 from the reference is applied to the sea salt surrogate (NaCl$^+$) to estimate the sea salt mass concentration in this study.

### 3.1.2 Chemical composition of the aerosol over the Atlantic

Based on the data quality control in steps above, an overview of the chemical composition of Atlantic aerosol particles during 4 cruises is given by Figure 2, including AMS default species (organics, sulfate, nitrate and ammonium), estimated marine products MSA and sea-salt, as well as BC. Since the ship tracks on latitude are nearly monotonic to the cruise time, particle chemical composition as a function of latitude gives both temporal variation and latitude distribution. Air mass back trajectories indicate that the air masses captured by the aerosol container during Polarstern cruises came either from the

ocean or were influenced by the continents (Europe, Africa or South America). These two air mass categories, marine air mass and mixed air mass with continental influence, are also marked in Figure 2. The classification into five air mass groups (three for marine, two with continental influence) is briefly described in Table S1, which is consistent with those in previous on-board studies over the Atlantic Ocean (Maßling et al., 2003; Norman and Leck, 2005; Virkkula, Aki et al., 2006; Virkkula, A. et al., 2006). Additionally, the statistics of seasonal variation of PM$_1$ chemical composition over the North (>5 °,

i.e. 5 °N) and South (<-5 °, i.e. 5 °S) of the Atlantic are provided in Table 3. Since there is no clear seasonal difference of the chemical composition of measured PM$_1$ in the area near the equator, the average, median, standard deviation and percentage in total were calculated for the region from 5 °S to 5 °N (-5 ° ~ 5 °) on latitude (defined as tropic in this study) regardless the season, which are also shown in Table 3.

Over all measurements, the total measured submicron particle mass concentration varies over a large range from 0.22 μg m$^{-3}$

to 14.15 μg m$^{-3}$. Its median is 1.83 μg m$^{-3}$, between the clean marine case (from 0.27 to 1.05 μg m$^{-3}$) measured over the Northeast Atlantic (Ovadnevaite et al., 2014), and the case of mixed marine-continental air masses (from 3.69 to 4.17 μg m$^{-3}$) over the North Atlantic (Dall'Osto et al., 2010). This suggests that the detected aerosol during the Polarstern cruises was from mixed sources as hinted by the air mass origins. On average, the measured particle mass during 4 cruises is contributed half by sulfate (50 ± 13 %), which is followed by organics (21 ± 9 %), sea salt (12 ± 11 %), ammonium (9 ± 4 %), BC (5 ±

4%) and nitrate (3 ± 2%). However, the mass concentrations of total particle as well as individual species show large variation associated with the time and obviously they are not distributed normally. Hence, the median rather than average of the mass concentration is used below.



During all 4 cruises, sulfate dominated the particle mass most of time, showing the highest median of mass concentration (0.78 µg m⁻³) in all six species. It could contribute up to 85 % to the total particle mass concentration. Seasonal discrepancy is found for sulfate mass concentration (higher level in spring than in autumn), especially over the South Atlantic (Table 3). Similar seasonal variation of the marine biogenic tracer MSA was also observed (Huang et al., 2017), suggesting the

biogenic sources contributed significantly to sulfate. Organics (here including MSA) are the second most abundant species (median: 0.26 µg m⁻³). The dominance of sulfate and organic is similar to other observations over the Atlantic Ocean (Dall'Osto et al., 2010; Zorn et al., 2008). Different from sulfate and MSA which have clear seasonal pattern, organics did not show pronounced variation associated with season, but more with location. The mass concentration of organics elevated with air masses under continental influence. During CR2 and CR4, organics even dominated the total measured particle mass

concentration (up to 59 %) in the region between 0 ° and 15 °N. Meanwhile, the non-marine species BC, despite tiny contribution to the total particle mass, was tending to increase together with organics. It indicates that anthropogenic emissions may be a significant contributor to the organics in this part of the Atlantic Ocean.

The marine primary product sea salt is found to play a minor role in the measured submicron particles during Polarstern cruises. However, associated with the elevated wind speed, the mass concentration of sea salt can reach 1.63 µg m⁻³ while

taking up to 66 % of the total particle mass loading (Figure 2). It is still comparable to the sea salt's importance reported in winter over the Northeast Atlantic (66-84%) (Ovadnevaite et al., 2014) and in the sub-Antarctic island (47%) (Schmale et al., 2013a) when winds were high.

The ammonium concentrations didn´t follow a clear seasonal trend, although its precursor ammonia could be emitted from ocean (Ikeda, 2014; Johnson et al., 2008). The absence in seasonality suggests particulate ammonium during Polarstern

cruises was contributed by both anthropogenic and biogenic sources. Nitrate also showed no seasonal pattern during all cruises and low mass concentration (0.04 µg m⁻³). The absence of nitrate is not surprising, because on the one hand it has no marine sources, and on the other hand, nitrate would mainly occur in supermicron particles. That is because nitrate would react with sulfuric acid and escape from submicron particles in form of gaseous nitric acid which will relocate to larger particles, e.g. sea salt particles (Saltzman, 2009).

**3.2 Source apportionment for organics aerosol (OA)**

Source identification and apportionment for the MBL aerosol particles are necessary for better understanding of their chemical characteristics and transformations. Organics are one of the main constituents of aerosol particles and contain hundreds of compounds from various origins. Analysing the source of organics may provide hints for evaluating origins and processes of other components. For the source apportionment of organics, PMF analysis was performed on the dataset of

high-resolution organic aerosol (OA) mass spectra following the construction from Ulbrich et al. (2009). Based on a comprehensive evaluation of different number of factors (regarding both time series and mass spectral pattern) and the rotational forcing parameters fPeak, a 5-factor solution was selected with fPeak =0 and Q/Qₑₓₚ = 0.9246. A summary of validation of the model and selection of the five-factor PMF solution is provided in the Figure S7. The mass spectra and time





series of the chosen solution did not change pronouncedly with the different fPeak as well as different seed (Figure S7). For the 4-factor solution, the nitrogen-containing OA (NOA) factor was missing, while oxygenated OA (OOA) factor split into two factors in the 6-factor solution (Figure S8). More details of both mass spectral profile and time series of 4- and 6-factor solutions can be found in Figure S8. To sum up, the 5-factor solution can properly explain the total OA with physical

meanings for each factor.

Among the five factors, four secondary (SOA) and one primary (POA) organic aerosol particle components were found in the Polarstern measurements. Their temporal variation and mass spectral profiles are shown in Figure 3. The SOA components include: (1) marine organic aerosol (MOA) with the highest S/C ratio (0.030), correlated well with the marine tracer MSA ($R^2$ = 0.83). These S/C ratios derived from the  PMF analysis tool contain however certain estimation

uncertainties and have therefore  to be used with caution (Aiken et al., 2007); (2) nitrogen-containing organic aerosol (NOA) with the highest N/C ratio (0.124); (3) oxygenated organic aerosol (OOA) correlated with continental tracer $NO_3$ ($R^2$ = 0.52); (4) aged primary organic aerosol (aPOA) with highest O/C ratio (1.35), varying temporally with BC ($R^2$=0.68). The only primary organic aerosol (POA) factor is characterized by abundant $C_xH_y$ ions consequently with lowest O/C ratio (0.23) of all five factors.

Figure 4 shows (a) the mass fraction of all 5 factors in total measured organics, as well as the functional groups composition and (b) the diurnal variation of each factor. Note that the mass fraction pie represents the average mass fraction of each factor derived from PMF analysis, and the "residuals" part of 0.4% is ignored in the pie chart but existing. The residuals correspond to the unexplained organics by PMF. Table 4 summarizes all comparisons for the identification of the five OA factors, including: (1) correlations of time series between the OA factors and the measured tracers; and (2) correlations of

mass spectral pattern between the OA factors and the identified OA sources in the literatures. Details on characteristics of each OA component, including mass spectral profile, temporal variation, and associations with different sources and processes are discussed in the following sections.

### 3.2.1 Marine organic aerosol (MOA)

The MOA factor is well correlated with the marine tracer MSA ($R^2$ = 0.83, Figure 3); it consequently can be linked to

oxidation of DMS emitted by phytoplankton. One characteristic of the MOA factor is a high contribution from $C_xS_y^+$ ions (7 %), which include mainly MSA identified ions $CH_3SO_2^+$ (*m/z* 78.985), $CH_2SO_2^+$ (*m/z* 77.978), $CH_4SO_3^+$ (*m/z* 95.988), $CHS^+$ (*m/z* 44.980), and $CH_2S^+$ (*m/z* 45.988). This results in a high S/C ratio (0.030), which is 10 to 30 times higher than that of other factors (Figure 3). Despite the previously mentioned caution, the S/C ratios can still provide indication on significance of sulfur when calculated with the same tool.  The S/C ratio of the MOA factor is also over twice that of marine

factor observed in Paris (0.013, Crippa et al., 2013b), implying a stronger influence from marine phytoplankton on aerosol particles over the ocean than those in the coast city. Moreover, the mass spectral pattern of the MOA factor is in positive agreement with reported marine origin factors (Table 4).



The MOA factor contributes averagely 16% to total OA mass, with the median of mass concentration as 0.04 μg m$^{-3}$. The enhancement of MOA mass concentration was observed to be independent on the air mass categories and most MOA peaks were in association with marine air mass. This supports the MOA is mainly from the ocean. Although organic sulfur species play a remarkable role, oxygenated organic ions are still the major species, accounting for 52 % of the MOA mass loading

($C_xH_yO_1^+$ 30 %, $C_xH_yO_z^+$ 22 %, Figure 4a). They are followed by hydrocarbon ions, taking 30 % of the MOA mass loading. $CH_3^+$ is the most abundant ion of $C_xH_y^+$ family, contributing 43% to the total $C_xH_y^+$ group mass loading in the MOA. As a result, MOA shows highest H/C ratio (1.73) of all five OA factors, similar to those reported previously: 1.57 (MOA) in Paris summer (Crippa et al., 2013b) and 1.8 (MOOA) in the Bird Island, Sub-Antarctic (Schmale et al., 2013a). This suggests that high level of hydrocarbon ions in particular $CH_3^+$ together with $C_xS_y^+$ ions could be an important characteristic

for marine source SOA.

Comparing to the standard MSA mass spectral profile (only organics signals), MOA includes almost all organic ions observed in pure MSA and some extra oxygenated ions which are absent or negligible in MSA. It indicates that the MOA factor consists of not only MSA but also other organic components from the same source or process as MSA. This is also proved by the relationship between mass concentration of MOA and MSA (slope = 2.19, R$^2$ =0.83). Significant oxygen-

15 contained ions such as $CO_2^+$ (m/z 43.990) and $COOH^+$ (m/z 44.998) could be related to carboxylic acids which are an important composition of secondary marine aerosol (Decesari et al., 2011; Fu et al., 2011; Fu et al., 2013). The plausible contributors could be isoprene and monoterpenes emitted by marine phytoplankton, which can be easily oxidized by OH radical and NO$_3$ radical to form highly oxygenated products such as 2-methylglyceric acid, 2-methyltetrol and pinic acids (Claeys et al., 2010; Fu et al., 2011; Hu et al., 2013). Marine-origin isoprene and monoterpenes could be the precursors of

20 carboxylic acid as well as organosulfate in SOA (Claeys et al., 2010; Fu et al., 2011; Hu et al., 2013; Iinuma et al., 2007; Surratt et al., 2008; Surratt et al., 2007). They may offer $CO_2^+$ and $C_xS_j^+$ fragments in the MOA factor. As shown in Figure 4, the diurnal cycle of the MOA suggests photo-oxidation is a main process. It shows a small but clear elevation in the afternoon, reaching the maximum (0.05 μg m$^{-3}$) at 16:00 almost when the global radiation started declining, which reflects the accumulation of photochemical production. The minimum of the diurnal variation (0.04 μg m$^{-3}$) appears around 09:00,

probably linking to the increase of mixing layer in morning.

The diurnal cycle of MOA might have been weakened by averaging because the biological activities in autumn are usually lower than in spring. Thus, a "MOA dominating period" is selected for a case study (about 2 consecutive days from 19:40, 18.11.2012 to 04:20, 21.11.2012). As shown in Figure 5, the MOA plays important role in the total OA during the selected period, taking averagely 78% (up to 100%) of the total OA mass concentration. The MOA has consistent variation with

30 MSA in time series, resulting in quite stable MSA/MOA ratio of 52% ±9%. Also, NO$_3$, NH$_4$ and BC showed low mass concentrations (median: 0.04 μg m$^{-3}$, 0.08 μg m$^{-3}$ and 0.05 μg m$^{-3}$ respectively) close to their DLs, indicating negligible impact from anthropogenic emissions during this period. The diurnal pattern for this specific period, with minimum of 0.11 μg m$^{-3}$ at 07:00 and maximum of 0.25 μg m$^{-3}$ at 16:00, is more noticeable than the average case (Figure 4). Similar diurnal cycles are observed for MSA and sulfate, suggesting that MOA, MSA and sulfate are formed via the same secondary pathway.



Model studies found that the DMS is mainly (84% globally) removed via the photo-oxidation by OH radical (Kloster et al., 2006). Also, the aqueous-phase oxidation of DMS is dominant by $O_3$ during the cloud process, and yield significantly amount of MSA (Hoffmann et al., 2016). These findings support that the DMS oxidation is controlled by photochemical process and its products should show daytime maximum associated with the solar radiation. Besides, isoprene and monoterpenes can also react with oxidants especially OH radical to form oxidation products (Claeys et al., 2004), consequently contributing to the this OA component.

Since the MOA component is successfully traced by MSA, the relationship between the MSA and MOA should be applicable to estimate DMS-related organic aerosols over the Atlantic Ocean. The correlations between MSA and MOA in spring, autumn and tropic (with unclear seasonal variation, 5 ˚N to 5 ˚S of latitude) are shown in Figure 6. The scattering points are fitted using linear orthogonal distance regression (ODR). Overall, the correlations between MSA and MOA in three cases are robust ($R^2$ = 0.85, 0.53, 0.88), and consistent (slope = 0.57, 0.56, 0.56) when not considering the spatial difference. We infer that the relation between MSA and its concomitant secondary organic components (MOA in this study) is generally stable, only showing tiny discrepancy in different regions. In spring, the slope is slightly smaller in the South Atlantic (0.52) than in the North Atlantic (0.66), with excellent correlations in both hemispheres ($R^2$ = 0.87 ~ 0.92). In tropical region, the MSA-MOA relationship is similar as that in the average case in spring, but both components show lower level (MSA <0.1 μg m$^{-3}$, MOA <0.16 μg m$^{-3}$) than those in spring (MSA < 0.2 μg m$^{-3}$, MOA < 0.32 μg m$^{-3}$). The lowest amount of both MSA and MOA was observed in autumn probably due to the low biological activities. This also weakens the linear correlation especially in the south hemisphere ($R^2$ = 0.22). On average, a slope of 0.56 between MSA and MOA is obtained for the whole Atlantic. Accordingly, the MOA mass concentration could be estimated as the production of the MSA concentration time the factor (i.e., 1.79 for average, 1.52 for the North Atlantic while 1.92 for the South). This can be useful for better estimation of marine DMS related SOA both in field measurements and in models.

### 3.2.2 Nitrogen-containing organic aerosol (NOA)

The nitrogen-containing OA (NOA) component showed a unique time series with poor correlation with other four factors (all $R^2$ are below 0.13). It varied up to 0.47 μg m$^{-3}$, but with median of 0.03 μg m$^{-3}$ which is quite close to the detection limit of organics. Similar to the MOA, the variation of NOA mass concentration shows independence of the different air masses, suggesting that the NOA may be of an oceanic origin. The NOA is characterized by remarkable contribution from organonitrogen (ON), mainly $C_xH_yN^+$ fragments (17% of the total NOA mass loading, Figure 4), showing the highest N/C atom ratio (0.124, Figure 3). Nevertheless, the NOA factor is dominated by oxygenated fragments including $C_xH_yO^+$ (30 %) and $C_xH_yO_z^+$ (28 %, Figure 4). The diurnal variation of NOA shows clear peak in the afternoon, reaching the maximum while the global radiation starts decreasing (Figure 4), indicating that the NOA factor is certainly composed of secondary organic products.

Nitrogen-containing OA component has been identified in several previous studies using AMS-PMF method, related to various sources highly dependent on the local situation, for example, Gentoo penguins hatching activities (Schmale et al.,



2013b), local coffee roastery (Carbone et al., 2013), and local primary (industry) emissions (Aiken et al., 2009). In MBL, gaseous amines emitted by marine phytoplankton have been recognized as an important SOA source (Dall'Osto et al., 2012; Facchini et al., 2008; Jickells et al., 2013; Müller et al., 2009). This is because low molecular weight aliphatic amines such as methylamine ($CH_5N$), dimethylamine ($C_2H_7N$) and trimethylamine ($C_3H_9N$) can be excreted by marine organisms as

products of metabolic processes (Ge et al., 2011; Wang and Lee, 1994). During the Polarstern measurements, the NOA factor is well correlated with ON fragments, e.g. $C_2H_7N^+$ ($R^2 = 0.86$), $C_2H_6N^+$ ($R^2 = 0.77$), and $CH_4N^+$ ($R^2 = 0.65$), which could be generated from amines (McLafferty and Tureček, 1993). The N/C ratio of the NOA factor (0.124) is close to the one used in a global 3D chemistry-transport model in the paper from Kanakidou et al. (2012) to trace the ON from ocean sources (0.15) and different from the ratio for BB and anthropogenic sources (0.3). The NOA factor also shows similarity

($R^2 = 0.70$) of mass spectral profile as that reported by Sun et al. (2011) who found the NOA factor in the New York City might be related to marine emissions and local industry emissions. The characteristic ions such as $C_2H_7N^+$ and $C_2H_6N^+$ may be attributed to the dimethylamine ($C_2H_7N$), which has been found as characteristic and abundant amines in marine emissions in previous studies (Facchini et al., 2008; Gibb et al., 1999; Müller et al., 2009).

Meanwhile, the NOA is neither correlated with BC ($R^2 = 0.17$) nor $NO_3$ ($R^2 = 0.06$), excluding the possibility of combustion

and anthropogenic (continental) sources. This factor shows different temporal variation from MSA ($R^2 = 0.01$) as well. However, this is consistent with findings in the previous marine measurements which related the NOA to biogenic amines (Facchini et al., 2008; Miyazaki et al., 2011; Müller et al., 2009). Moreover, the NOA contains two characteristic makers for amino acids, i.e. $CH_4N^+$ (*m/z* 30) and $C_2H_4N^+$ (*m/z* 42) recognized by Schneider et al. (2011), although the NOA shows poor similarity of mass spectra (partly shown in Table 4) to 15 pure amino acids (Schneider et al., 2011). The possibility cannot

be completely excluded that amino acids (proteins) from marine primary organic components (Hawkins and Russell, 2010) are transformed (e.g. photo-oxidized, or decarboxylated) and contribute to this factor. Although the NOA mass concentration has similar trend with the water temperature (Figure S9), the reason for this coincidence is still unclear. It might be related to the positive effect of temperature on metabolism rate including N excretion of marine microorganisms (Alcaraz et al., 2013; Ikeda, 1985; Ikeda, 2014; López-Urrutia et al., 2006). Therefore, we speculate that the NOA component is mainly related to

marine amines emitted from phytoplankton via gas-to-particle conversion. The contribution of amino acids (proteins) is not recognized but cannot be fully ruled out.

### 3.2.3 Oxygenated organic aerosol (OOA)

Though measuring over the ocean, 19 % of organic aerosol mass was contributed by the long-range transported emissions, presented by the OOA factor with median of 0.04 μg m$^{-3}$ for mass concentration. The OOA factor was identified by

anthropogenic tracer $NO_3$ regarding time series ($R^2 = 0.52$, Figure 3) and resemblance of mass spectral profile to the reported continental OOA factors (Table 4). For instance, the OOA in this study is in excellent agreement ($R^2 = 0.98$) with a continental organics factor observed over the central Arctic Ocean (Chang et al., 2011). It is also similar ($R^2 = 0.67$) to the average OOA factor based on nine urban measurements (Ng et al., 2011). The OOA factor is dominated by $C_xH_yO^+$



fragments (contributing 40 % to the total factor mass), followed by $C_xH_yO_z^+$ (26 %) as shown in Figure 4. Significant contributions are observed from $CO_2^+$ (*m/z* 44, 21 % of total OOA mass loading) and $C_2H_3O^+$ (*m/z* 43, 6 %), comparable with values in previous studies in urban or rural areas (e.g. Lanz et al., 2007; Ng et al., 2011; Poulain et al., 2011a). Another carboxylic acid fragment $CO^+$ (m/z 28, 21 % of total OOA mass loading) is estimated according to $CO_2^+$ ion, and thus $CO^+$ is

not particularly discussed in the present work. There is no clear diurnal pattern for the OOA factor (Figure 4). The OOA mass concentration dropped in the early morning from 05:00, reaching minimum at ~10:00 then being stable until midnight. This may be explained by the rising mixing layer in the morning which dilutes the particle concentration, and/or increasing temperature after sunrise drives volatile species from particles into gas phase. Hence, the OOA is not likely to be contributed by locally photochemical formation. Moreover, as shown in Figure 3, significant elevation of OOA and $NO_3$ mass

concentrations (up to 2.70 μg m$^{-3}$ and 0.45 μg m$^{-3}$) was mainly associated with the mixed air mass with continental influence especially when close to Africa and Europe. This also supports the continental outflow is the main source of the OOA factor.

### 3.2.4 Aged primary organic aerosol (aPOA)

Of all five factors, the aPOA is the only one correlated with BC ($R^2 = 0.69$, Figure 3). It is the most abundant component of organic aerosols, contributing 30 % of total OA mass. The median of aPOA mass concentration is 0.07 μg m$^{-3}$ (<DL to 1.38

15   μg m$^{-3}$). Similar to the OOA, the aPOA factor peaks usually appeared together with continental influence air masses, pointing to non-marine sources. One significant characteristic of the aPOA factor is its highly oxygenated level. Oxygenated ions including $C_xH_yO^+$ and $C_xH_yO_z^+$ account for 73 % of the total aPOA mass (Figure 4), hence aPOA exhibits the highest O/C ratio (O/C = 1.35), and lowest H/C ratio (H/C = 0.94) of all the OA factors.

Since correlated with combustion tracer BC, this factor can be considered to connect with various burning sources and

processes, such as ship exhausts, cooking, and burning with biomass. Given that the influence of direct ship emissions from Polarstern (including the kitchen exhausts using the same chimney) has been removed, the own-ship exhausts as well as cooking emissions could be left out of consideration. Moreover, the mass spectral profile of aPOA factor shows no similarity with any fresh combustion emissions, including 1) the fresh ship emissions factor ($R^2 = 0.00$) (Chang et al., 2011) , 2) the average BB organic aerosols BBOA (Ng et al., 2011) with $R^2 = 0.09$, and 3) the average HOA factor ($R^2 = 0.01$) (Ng et al.,

2011), supporting the conclusion above. Because most of the time Polarstern travelled along the main water way (Figure S10), a likely source of aPOA could be the transported ship exhausts from other ships. Sage et al. (2008) found that SOA could be quickly formed from fresh diesel exhausts, and the aged products were increasingly oxidized with time resembling the profile of atmospheric aged OA within a few hours of oxidation. The aPOA factor in this study has very similar mass spectrum ($R^2 = 0.73$) to the OA below clouds measured by a flight impacted by ship emissions (Coggon et al., 2012). The

aPOA mass spectral profile is also correlated with the photo-oxidation products of diluted diesel generator exhaust ($R^2 = 0.46$) (Sage et al., 2008). Thus, the aged ship emission should be one of the important contributors of the aPOA component. The mass concentration of aPOA increased when the ship was in the English Channel where the marine traffic is very busy (Figure S10).



The ship traffic, however, can hardly explain the remarkable seasonal elevation of aPOA in November near the equator. Thus, the other plausible source of the aPOA, the aged BB emissions, should be considered. It is well known that Africa is the biggest single continental source of the BB emissions with strong seasonality (Cooke et al., 1996; Giglio et al., 2006; Roberts et al., 2009). The fire maps (Figure S11) showed that in the west and middle Africa, there were more intensive fire

points occurred in November than in April/May, consistent with higher mass concentration of aPOA in the CR2 and CR4 (both covering the November) than other two cruises. In this study, the aPOA factor contains extremely high oxygenated ions (especially m/z 44, $CO_2^+$, its relative amount to the total aPOA mass, i.e. $f_{44}$, is 32%), while nearly no BB tracer ions at m/z 60 and m/z 73 (Bertrand et al., 2017; Capes et al., 2008; Cubison et al., 2011), i.e., $C_2H_4O_2^+$ ($f_{60}$ = 0.06%) and $C_3H_5O_2^+$ ($f_{73}$ = 0%). The $f_{60}$ in the aPOA factor is much lower than the background level without apparent BB influence (0.3% $\pm$

0.06%) provided by Cubison et al. (2011); but it increased up to 0.18% during the high aPOA period when the aPOA average mass concentration and mass fraction (in total OA) reached 1.02 $\pm$ 0.14 μg m$^{-3}$ and 47% $\pm$ 3%. The absence of these BB tracer ions could be reasonable if taking the aging process during long-range transport into the consideration. Fresh BB emissions can be quickly photochemically aged (in several hours), resulting in formation of the new OA with higher degree of oxygenation and a significant decrease of BB tracers as well as saturated hydrocarbon compounds. This has been

observed in both laboratory (Bertrand et al., 2017; Bertrand et al., 2018; Grieshop et al., 2009a; Grieshop et al., 2009b) and field measurements (Capes et al., 2008; DeCarlo et al., 2010). During the Polarstern cruises, the BB tracers in aPOA factor decayed severely, possibly due to the chemical aging in long-range (i.e. long-period) transport and dilution effect of clean marine air masses.

Another evidence for aged BB emissions is: the aPOA is well correlated with $C_xH_yO_zN_w^+$ fragments (from AMS, $R^2$ = 0.83)

and potassium ion (K$^+$, from offline measurements, $R^2$ = 0.61) as shown in Figure 7. The $C_xH_yO_zN_w^+$ ions have been related to the photo-oxidation product of m-cresol, a typical wood burning emission (Iinuma et al., 2010; Poulain et al., 2011a), while potassium is commonly regarded as an unreactive tracer of BB emissions. In addition, the particles measured in the range from ~15 °N to 15 °S (close to the west and middle Africa) showed external mixing state by the HTDMA measurements (details in an accompany paper by Wu et al., in preparation). It is the typical property of BB emissions.

Therefore, aPOA is possibly contributed by aged combustion emissions from both ship traffic and biomass burning. Aged BB emissions may be the dominant source, leading to a spatial and seasonal dependence.

### 3.2.5 Primary organic aerosol (POA)

The only true identified primary factor, POA, is related to marine primary emissions. It is characterized by high contribution of hydrocarbon ions (64 % of total factor mass concentration, Figure 4), which is usually considered as a feature of primary

emissions. Also, the diurnal profile of POA (Figure 4) indicates there is no photo-oxidation contribution to this factor. The time series of POA mass concentration is spiky and varies from <DL to 0.67 μg m$^{-3}$ with the median of 0.04 μg m$^{-3}$. The POA shows no synchronicity with continental influenced air masses, implying plausibly marine sources for the POA. It averagely contributed 19% to the total OA mass, comparable to two marine SOA factors (MOA 16 %, NOA 16 %). Since



the ship contamination periods have been eliminated and the POA is not correlated with BC ($R^2 = 0.04$), fresh ship emissions are not likely to be a contributor to this primary factor. The primary emissions from the ocean are considered as a main source because of the similarity ($R^2 = 0.61$) on mass spectral profile between this factor and primary marine organic aerosol during high biological activities in Mace Head Ireland (Ovadnevaite et al., 2011). Moreover, CHN fragments take pronounced portion (5%, Figure 4) in total POA mass loadings, only smaller than its portion in the NOA factor. They may be related to amines which have been observed in primary marine organic aerosols in recent measurements (Frossard et al., 2014; Quinn et al., 2014; Quinn et al., 2015).

There are one fourth of POA mass identified to be oxygenated organic compounds ($C_xH_yO^+$, $C_xH_yO_z^+$). It is worth noting that carboxylic acids can also be supplied via primary pathway and have been identified as a distinct type of marine primary organic matters (Hawkins and Russell, 2010). Sea water is a pool of dissolved organic matters contributed by direct algal release, metabolism of marine heterotrophs, bacterial and viral lysis, and cell senescence (Hansell and Carlson, 2002). The sea spray aerosol is mechanically generated from the bubble bursting processes, in which the chemoselective transfer occurs and results in the enrichment of organic compounds in the primary marine aerosol particles (Aller et al., 2005; Kuznetsova et al., 2005; Russell et al., 2010; Schmitt-Kopplin et al., 2012).

Overall, the POA is not correlated with the sea salt ($R^2 = 0.01$), nether the wind speed ($R^2 = 0.01$), even if similar variation trend was found in several periods, such as the south parts (from 33 °S to 0 °) in CR2 and CR4, and the middle part (5 °N to 20 °N) in CR3 (Figure S12). However, whether the primary marine organics should be proportional to the sea salt is still questionable. On one hand, several studies observed that the production mass flux of the sea salt exponentially enhanced with increasing surface wind speed, and the strong wind usually produces the sea spray particles in the supermicrometer size range (Grythe et al., 2014; Ovadnevaite et al., 2012). On the other hand, the mass fraction of organics in the sea spray aerosol was found to increase with the decreasing particle size (Gantt et al., 2011; Quinn et al., 2015). This suggests that the ocean-source primary organic aerosol could be not correlated with the sea salt. Due to the resemblance between mass spectral profile of POA and reported primary marine organic aerosol, we speculate that the POA factor could be attributed to the mechanically generated ocean products.

## 4 Summary and conclusions

In this study, chemical composition of the submicron aerosols was investigated based on a unique dataset during 4 open-ocean cruises over the Atlantic in 2011 and 2012. PM$_1$ composition varies dynamically during the cruises, averagely composed of sulfate (50%), organics (21%), sea salt (12%), ammonium (9%), BC (5%) and nitrate (3%). Organics are an important constituent of PM$_1$ and contain a large number of compounds from different sources. The PMF analysis was therefore performed to the high-resolution mass spectra matrices of OA and identified five factors linked to the distinct sources. Figure 8 illustrates the latitudinal distribution of OA source contributions associated with the OA mass concentration over the Atlantic, which also gives a summary of OA source apportionment. Related the DMS oxidation, the



MOA factor shows prominent seasonality, higher contribution to the total OA mass in spring than in autumn; moreover, higher contribution over the South Atlantic than the North in spring. Especially, the MOA could be the exclusive contributor of the OA at around 16 ˚S in spring, linked to the high biological activity fuelled by the Benguela upwelling which is a northward flowing ocean current along the west coast of southern Africa from Cape Point (Nelson and Hutchings, 1983)

bringing the nutrients from the deep cold water. This seasonality is, however, not observed for the other two marine factors, i.e., the NOA factor with precursors of biogenic amines, and the POA from primary marine emissions. They both played a significant role in the clean regions with low particle mass concentration (e.g. in CR3 when the ship started from Punta Arenas). The continental influence is clearly recognized in Figure 8, represented by OOA and aPOA factors. Both factors took dominant even overwhelming mass fractions when close to the land. The impact of wild fires in the West and Middle

Africa to the open-ocean OA could be inferred from the significance of aPOA between 15 ˚N and 15 ˚S, where aPOA contributed more than 50% to the total OA (e.g. in CR2 and CR4).

The main findings of this study support that the sulfate is the dominant species of submicron aerosol particles over the open-ocean of Atlantic. Sulfate is significantly contributed by the biogenic sources (e.g. DMS) due to the noticeable seasonality of its mass concentration. The prominence of organics mass concentration has been found when the ship track was near the

15 Europe or Africa. This is mainly linked to the continental outflow including anthropogenic pollutants and biomass burning emissions which can bring abundant organic aerosols. Moreover, the maritime traffic density is higher when closer to the continents, so the ship emissions would also contribute to MBL organic aerosols. Despite the marked continental influence on organics over some regions of the Atlantic, the detailed source apportionment of OA displayed that oceanic OA is ubiquitous and, in a great part of cruising areas, dominates the total OA mass loadings. On average, marine sources

(represented by POA, MOA, and NOA) contributes 51% to the total OA mass concentration, nearly equal to that of the non-marine emissions (OOA and aPOA, 49%). One of the marine factors, MOA, is highly controlled by seasonal phytoplankton blooming (DMS production) and could even be the sole source of the OA in some regions with high biological activities, while the other two represent a background oceanic contribution and are more visible in the clean marine areas.

The results from this work also suggested that the South Atlantic is less polluted by the continental transport and human

activities (such as ship traffic) than the North Atlantic regarding the OA. Nevertheless, the OA could also be dominant by the marine sources in some regions over the North Atlantic, e.g. ~40˚N in CR3, while the opposite situation was found in CR1 in the same season. This shows that the discrepancy of OA source contributions in the same regions exists, requiring more future studies with multiple techniques as well as analysis methods. Finally, as a co-product of the source apportionment, a solid linear correlation has been found between MOA and MSA, which enables the estimation of marine SOA with DMS

origin in spring as MSA mass concentration times a factor of 1.79. This may be applicable in both field measurements and model study with a focus on the marine aerosol.



**Acknowledgements.** We thank all scientists and crews on board R/V Polarstern. We are grateful for the support from Alfred Wegener Institute (AWI) and Germany Weather Service (DWD) for sharing data of cruises, including data of navigation, oceanography, meteorology, and air mass back trajectories. We thank all the support through the following projects and research programs: (1) the Gottfried Wilhelm Leibniz Association (OCEANET project in the framework of PAKT), and (2) Polarstern expeditions AWI_ANT27/4, AWI_ANT28/1, AWI_ANT28/5, AWI_ANT29/1.

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





**Table 1. The description of 4 cruises**

| Expeditions | Starting point and destination | Duration | Season (in NH) |
|---|---|---|---|
| Cruise 1 (ANT-XXVII/4) | Cape Town - Bremerhaven | 20.04 -20.05.2011 | Spring |
| Cruise 2 (ANT-XXVIII/1) | Bremerhaven - Cape Town | 28.10 -01.12.2011 | Autumn |
| Cruise 3 (ANT-XXVIII/5) | Punta Arenas -Bremerhaven | 10.04 -15.05.2012 | Spring |
| Cruise 4 (ANT-XXIX/1) | Bremerhaven - Cape Town | 27.10 -27.11.2012 | Autumn |

**Table 2.  The 20-min detection limits during 4 Polarstern cruises ($\mu g\ m^{-3}$)**

|  | CR1 | CR2 | CR3 | CR4 |
|---|---|---|---|---|
| **Organics** | 0.024 | 0.025 | 0.017 | 0.022 |
| **Sulfate** | 0.011 | 0.018 | 0.014 | 0.012 |
| **Nitrate** | 0.004 | 0.006 | 0.008 | 0.005 |
| **Ammonium** | 0.021 | 0.020 | 0.021 | 0.018 |
| **Chloride** | 0.005 | 0.006 | 0.005 | 0.007 |
| **MSA** | 0.002 | 0.003 | 0.002 | 0.006 |



**Table 3. Seasonal chemical composition of measured PM$_1$ over the Atlantic (µg m$^{-3}$)**

| | | Spring | | | | Autumn | | | |
|---|---|---|---|---|---|---|---|---|---|
| | | average | median | σ | % | average | median | σ | % |
| North Atlantic (>5°) | SO$_4$ | 1.38 | 1.02 | 1.09 | 51% | 0.76 | 0.63 | 0.55 | 42% |
| | Org | 0.53 | 0.38 | 0.52 | 20% | 0.47 | 0.27 | 0.51 | 26% |
| | SS | 0.27 | 0.17 | 0.26 | 10% | 0.16 | 0.10 | 0.16 | 9% |
| | NH$_4$ | 0.29 | 0.19 | 0.31 | 11% | 0.20 | 0.15 | 0.15 | 11% |
| | NO$_3$ | 0.09 | 0.06 | 0.08 | 3% | 0.07 | 0.06 | 0.04 | 4% |
| | BC | 0.10 | 0.06 | 0.10 | 4% | 0.13 | 0.08 | 0.14 | 7% |
| | MSA | 0.04 | 0.03 | 0.03 | 1% | 0.01 | 0.01 | 0.01 | 1% |
| | Total | 2.71 | | | | 1.79 | | | |
| South Atlantic (<-5°) | SO$_4$ | 1.23 | 1.11 | 0.70 | 57% | 0.33 | 0.28 | 0.17 | 47% |
| | Org | 0.23 | 0.18 | 0.16 | 11% | 0.17 | 0.15 | 0.09 | 24% |
| | SS | 0.37 | 0.33 | 0.22 | 17% | 0.09 | 0.07 | 0.08 | 13% |
| | NH$_4$ | 0.15 | 0.08 | 0.12 | 7% | 0.07 | 0.07 | 0.04 | 10% |
| | NO$_3$ | 0.04 | 0.04 | 0.02 | 2% | 0.02 | 0.01 | 0.01 | 2% |
| | BC | 0.07 | 0.05 | 0.07 | 3% | 0.02 | 0.01 | 0.03 | 3% |
| | MSA | 0.05 | 0.04 | 0.04 | 2% | 0.01 | 0.01 | 0.00 | 1% |
| | Total | 2.14 | | | | 0.71 | | | |
| Tropic Atlantic (-5°~5°) | SO$_4$ | 1.03 | 0.93 | 0.65 | 50% | | | | |
| | Org | 0.46 | 0.34 | 0.33 | 23% | | | | |
| | SS | 0.13 | 0.12 | 0.08 | 6% | | | | |
| | NH$_4$ | 0.23 | 0.22 | 0.13 | 11% | | | | |
| | NO$_3$ | 0.04 | 0.03 | 0.02 | 2% | | | | |
| | BC | 0.14 | 0.10 | 0.12 | 7% | | | | |
| | MSA | 0.02 | 0.02 | 0.02 | 1% | | | | |
| | Total | 2.05 | | | | | | | |

(σ = standard deviation; SO$_4$ = sulfate, Org = organics, SS = sea salt, NH$_4$ = ammonium, NO$_3$ = nitrate)





**Table 4. Summary of correlations of time series and mass spectra for the five-factor PMF solution**

|  | MOA | NOA | OOA | aPOA | POA |
|---|---|---|---|---|---|
| Time series ($R^2$) | particulate MSA **(0.83)** | $C_2H_7N^+$ **(0.86)** | particulate $NO_3$ **(0.52)** | particulate BC **(0.68)** |  |
| Mass spectra ($R^2$) | MOOA[1] in Bird Island (Schmale et al., 2013a) **(0.71)** | NOA in New York (Sun et al., 2011) **(0.70)** | Continental organics (Chang et al., 2011) **(0.98)** | Beech smoldering (Weimer et al., 2008) **(0.68)** | Marine primary organic matter in Mace Head (Ovadnevaite et al., 2011) **(0.61)** |
|  | MOA in Paris summer (Crippa et al., 2013b) **(0.68)** | Alanine (Schneider et al., 2011) **(0.50)** | LVOOA[2] Paris winter (Crippa et al., 2013a) **(0.91)** | Fir smoldering (Weimer et al., 2008) **(0.68)** |  |
|  | Marine organics over Arctic (Chang et al., 2011) **(0.54)** |  | Average OOA (Ng et al., 2011) **(0.67)** | Oak smoldering (Weimer et al., 2008) **(0.62)** |  |
|  |  |  | Average LVOOA (Ng et al., 2011) **(0.66)** | Ship track (Coggon et al., 2012) **(0.40)** |  |

[1] MOOA: marine oxygenated organic aerosols

[2] LVOOA: low-volatility oxygenated organic aerosols





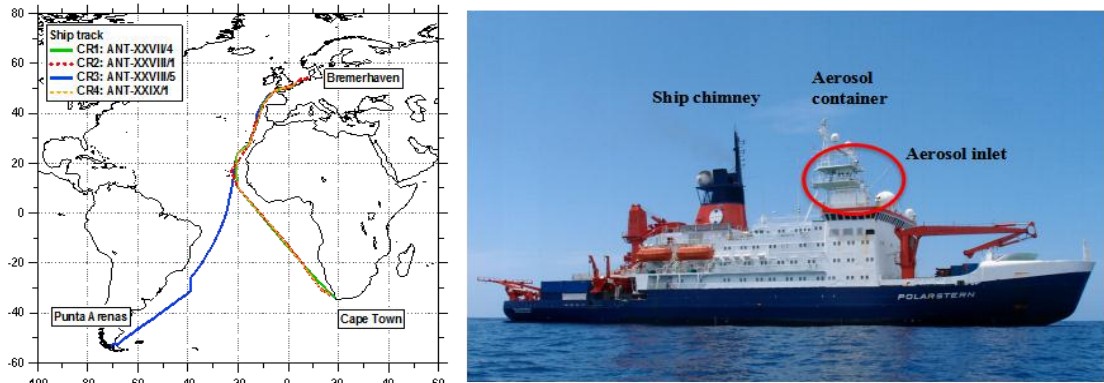

**Figure 1. (a) Ship tracks of 4 cruises; (b) The position of the container during Polarstern cruises.**

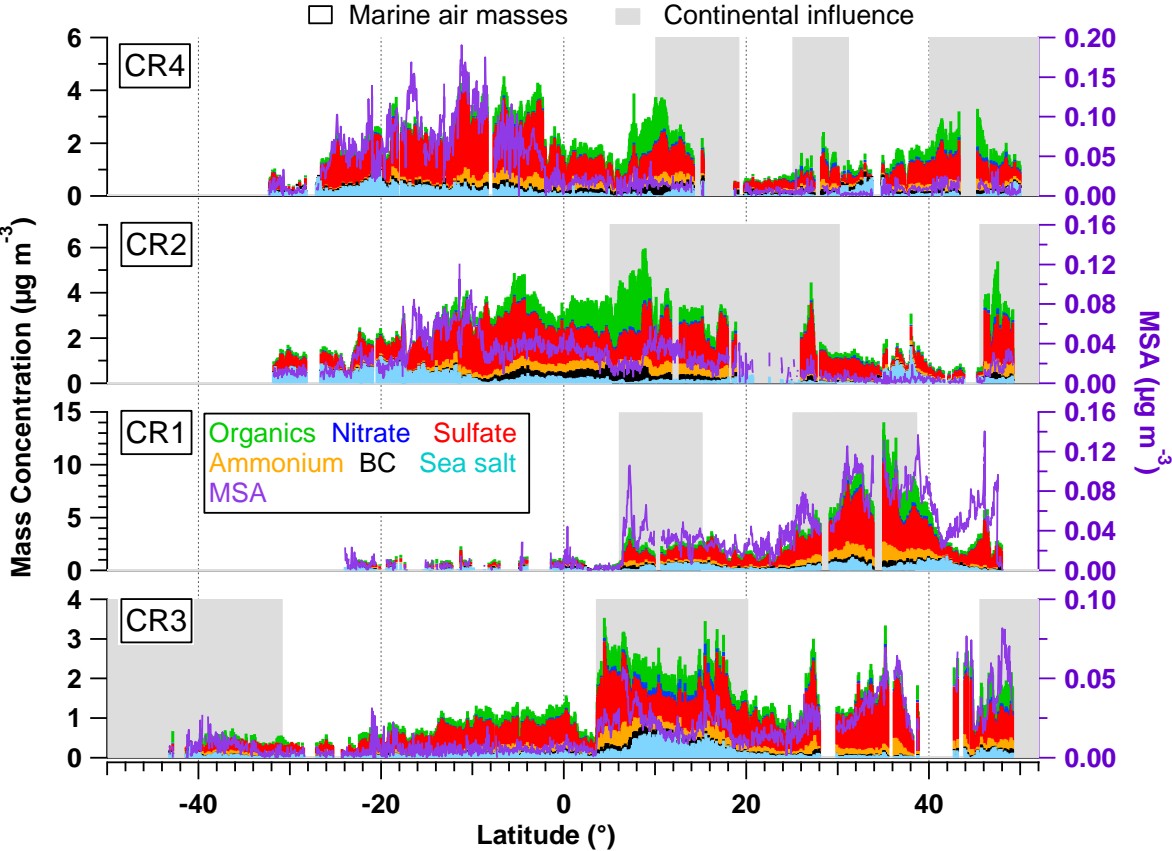

5   **Figure 2. Latitudinal variation of organics, nitrate, sulfate, ammonium, BC, sea salt (left axis) and MSA (right axis) in the four Polarstern cruises. Air masses with continental influence (grey) and originated from the ocean (blank) are marked on background. Note that CR1, CR2, and CR4 followed almost the same ship track between Bremerhaven and Cape Town, while the route of CR3 was different (from 15 °N) since starting from Punta Arenas.**





Figure 3. (a) High resolution mass spectra and (b) time series of 5 OA components. Also shown are simultaneous variation of tracer compounds on the right axes with marine (blank) and continental influenced (grey) air masses.



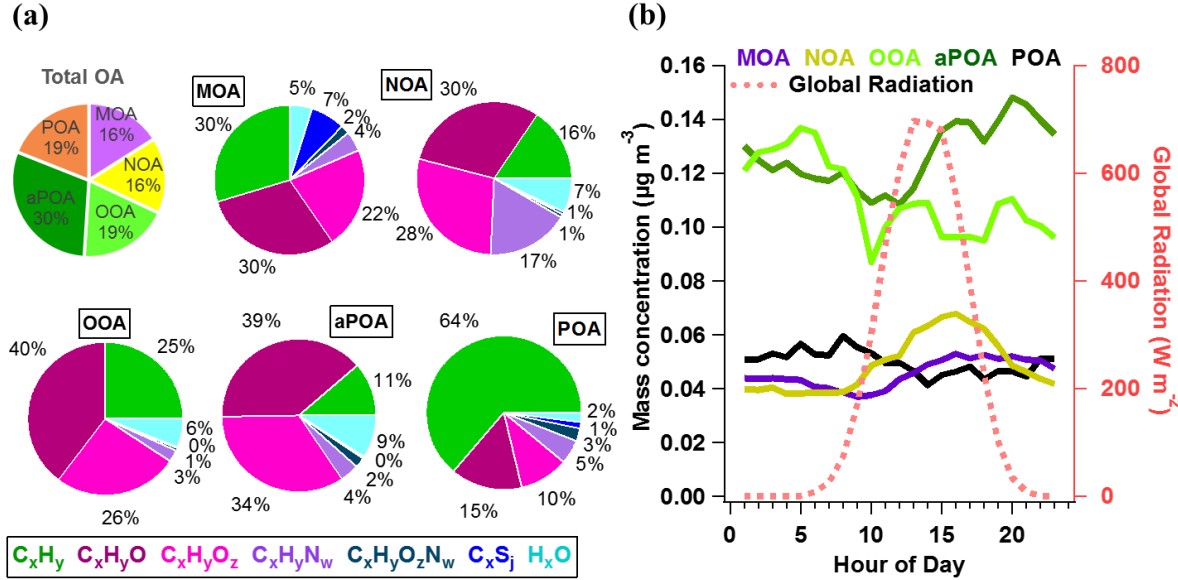

**Figure 4 (a) average mass fraction of each component in the total OA mass concentration, and the functional groups composition of each OA factor, and (b) diurnal variations of 5 OA factors with global radiation**

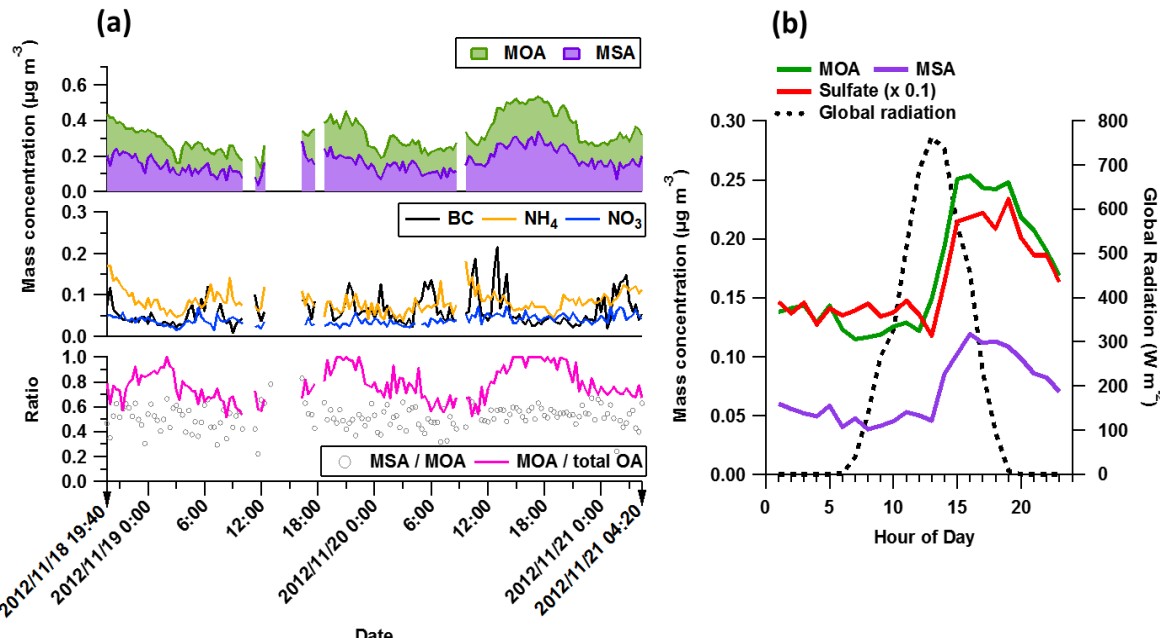

**Figure 5 The characteristics of the MOA dominating period: (a) time series of MOA and MSA (top), possibly non-marine species (BC, NH₄, NO₃; middle figure), as well as the ratio between MSA and MOA, and the mass fraction of MOA in the total OA (bottom); (b) the diurnal variation of MOA, MSA, Sulfate (reduced to one tenth for comparison) and the global radiation.**




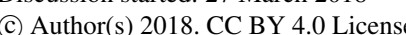

**Figure 6 Correlation between the MOA factor and MSA in spring and autumn over different regions of the Atlantic**

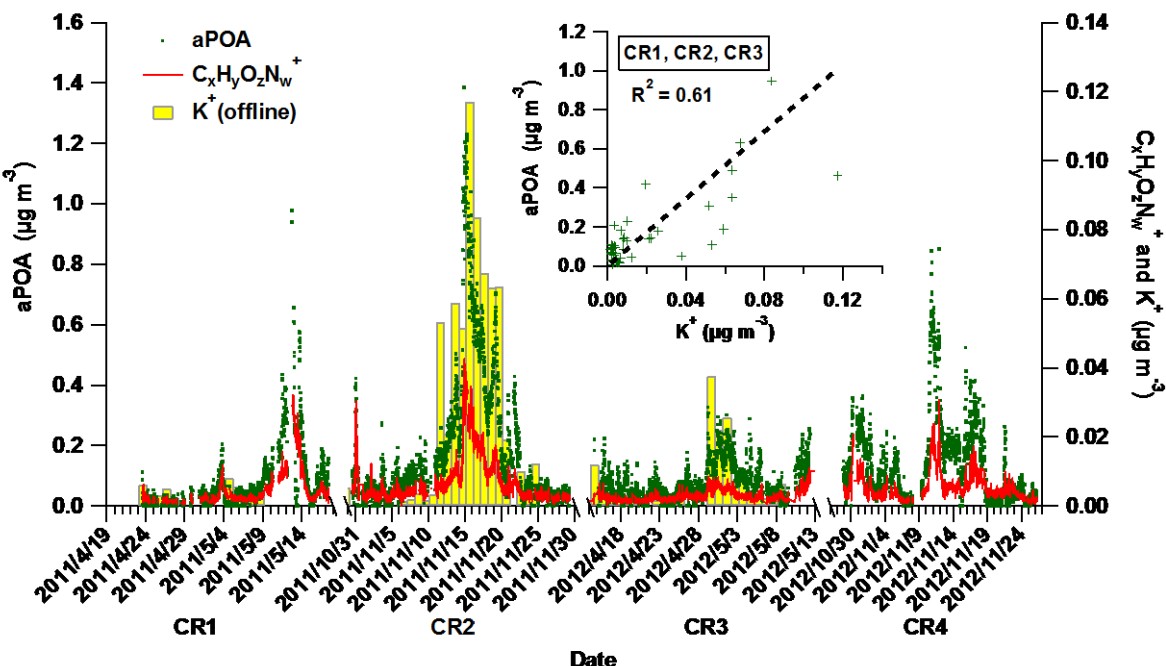

**Figure 7  The time-series of aPOA (green dots), $C_xH_yO_zN_w^+$ fragments (red line) and potassium ($K^+$, the yellow bar with grey box) during Polarstern cruises. Note that the daily mass concentration of $K^+$ is obtained from offline measurements only performed in the first three cruises. The scattering plot in the sub-window provides the correlation between daily average mass concentration of aPOA from AMS and potassium from offline measurements.**





**Figure 8 Latitude distribution of 5 OA factor mass fractions in the total OA with total OA mass concentration along the ship track during 4 cruises. The Benguela upwelling is marked in the figures for CR2 and CR4, while not displayed (but still exists) in CR1 and CR3.**

