# Peer review of "Source apportionment of the submicron organic aerosols over the Atlantic Ocean from 53 N to 53 S using HR-ToF-AMS"

_Atmospheric Chemistry and Physics, 2018_

## Referee Comment (RC1) · Anonymous Referee #1 · 24 Apr 2018

(My ratings for both scientific and presentation quality are between "Good" and "Fair", so I give one "Good" and one "Fair".)

General Comments:

The authors performed 4 month-long field measurements across 53°N-53°S over the Atlantic Ocean from 2011-2012 and reported convincing source apportionment based on adequate and well-processed datasets obtained from HR-ToF-AMS and other techniques. Due to few number of similar studies that covered long time series and/or oceanic regions, the findings in this paper certainly provide valuable insights into the sources and origins of marine submicron atmospheric aerosols. Overall, the paper dis-

cussed relevant scientific questions within the scope of ACP journal with novel methods and datasets, and the results are generally (but not all) supportive to the interpretations and conclusions, in spite of some technical concerns and vague presentation or expression that need to be further supported, examined, or re-phrased. Therefore, I would recommend this paper be accepted for publication once the following specific comments are completely addressed.

Suggestions for major improvements and revision:

1. The author(s) should have made best use of their valuable datasets as well as the previously published studies, and emphasized the significance of their findings, if they also agree that they haven't done this enough in the abstract and the in the introduction. Besides, the author(s) should also, on one hand, carefully refer to previous studies that used similar techniques for marine aerosol, and on the other hand, include necessary comparisons (if available) in their own discussions. For example: The authors should add proper references to the sentences ended in: Page 10 Line 5; Page 11, Line 25; Page 12, Line 27; Page 12, Line 34; Page 13, Line 6; Page 16, Line 24 (Might be useful: Charlson et al., 1987, Nature; Bonsang et al. 1992, GRL; Yassaa et al., 2008, Env. Chem.; Shaw, Gantt, and Meskhidze, 2010, Advances in Meteorology).

2. Conclusions discussing causality or reasoning must be carefully examined. Just give a few examples: Page 9, Line 5: The authors attributed "insufficient offline samples" to the weak correlation between AMS and offline sea salt. Actually this might not be a reasonable explanation especially if they used AMS data collected from the exactly same periods of time during the offline filter sampling. The data size itself should not affect the $R^2$, and the authors should also examine p-value of correlation for "meaningfulness". Furthermore, in this case, the authors should also clarify how they measured sea salts using the individual techniques and why they applied the method from Ovadnevaite et al. (2012). For example, what ions were included as sea salts? Did they count $Na^+$, $Cl^-$, $SO_4^{2-}$, $K^+$, $Mg^{2+}$, etc. in both? If NaCl accounted for different fraction from that in Ovadnevaite et al. (2012), was the scaling factor of 51 still suitable? Other-

wise, the "therefore" in Line 6, did not explain why the same scaling factor was applied, considering the correlation and the coverage of time ("full year measurements in the reference") discussed above was not supportive, or not relevant. Page 10, Line 18: The author stated "The ammonium concentrations didn't follow a clear seasonal trend, although its precursor ammonia could be emitted from ocean (Ikeda, 2014; Johnson et al., 2008). The absence in seasonality suggests particulate ammonium during Polarstern cruises was contributed by both anthropogenic and biogenic sources." The deduction did not support their conclusion. Page 13, Line 29: "The diurnal variation of NOA shows clear peak in the afternoon, reaching the maximum while the global radiation starts decreasing (Figure 4), indicating that the NOA factor is certainly composed of secondary organic products." The evidence is weak, and the authors should specify "global radiation" and cite papers that observed the similar diurnal trends of such a NOA factor, if available. Page 13, Line 20: The authors suggested "This can be useful for better estimation of marine DMS related SOA both in field measurements and in models". However, MSA as a fraction of SOA can vary largely and different from time to time (especially between summer and winter). In addition, MOA in this case might not be equivalent to SOA.

3. For better presentation quality and reading experience, the English language and scientific writing in this paper can be more precise and largely improved. Just give a few examples: Page 11, Line 9: "These S/C ratios derived from the PMF analysis tool contain however certain estimation uncertainties and have therefore to be used with caution." This seems to be a grammatically wrong sentence. Page 12, Line 24: "The minimum of the diurnal variation (0.04 $\mu$g m-3) appears around 09:00, probably linking to the increase of mixing layer in morning." This sentence needs to be re-phrased and also supported with references. Page 10, Line 27: I think it is more precise to say "57 hours" rather than "about 2 consecutive days", unless there was an interruption. Page 13, Line 6: "the this OA component". Despite the grammatical error and lack of references, "OA component" was vague in the context. Page 17, Line 17: In this paragraph, the author said "still questionable" and then "This suggests. . . could be not

correlated". This led to confusion due to the inappropriate English or logical expression.

Other technical and specific comments to be addressed:

1. Generally when discussing seasonality, the difference between "spring & autumn" might not be as distinct as that between "summer & winter", in term of many factors such as meteorological parameters and marine bioactivity. Besides the "spring vs. autumn" comparison, the authors may also want to look into "spring/autumn vs. tropic". In addition, their measurements on board was changing with time and location at the same time, so this will be different from those studies took place at a ground site over seasons. I wonder if the authors would like to make some comments on these.

2. The authors should try to clarify the influences from the "open oceans", "marine", and "coastal" when interpreting results in the discussions, even though the boundaries might be blurry. For example, on Page 11, Line 30, the author stated "The S/C ratio of the MOA factor is also over twice that of marine factor observed in Paris (0.013, Crippa et al., 2013b), implying a stronger influence from marine phytoplankton on aerosol particles over the ocean than those in the coast city.", but actually the abundance of phytoplankton can be much higher in the coastal areas. See https://earthobservatory.nasa.gov/GlobalMaps/view.php?d1=MY1DMM_CHLORA

3. The authors are suggested to add discussions for organosulfates, since they can make a considerable contribution to continental SOA masses at certain locations, and also derived from the same biogenic precursors over the oceans. For example, how is this class of compounds measured using AMS? Was it included in organics or sulfate, or neither?

4. Last but not least, the authors should revise the manuscript carefully by their own. Just give a few examples: 1) Page 12, Line 34: "Figure 4" – should this be Figure 5? 2) Acronym: define before use. For example, "SOA" was not defined but used in the abstract; "OA" was firstly defined on Page 10, Line 30 in the main text; "biomass burning" was defined but not used in many places. 3) Please be consistent when using

terms such as "fPeak" or "fpeak", "CxHyO" or "CxHyO1". 4) Please be consistent about adding a "_" between numerical values and their units. 5) Please specify "CxSj+" on Page 12, Line 21.

---

## Referee Comment (RC2) · Anonymous Referee #2 · 22 May 2018

The technical aspect of the current paper is very good, and the data of very high quality. Being able to collect so many cruises with HR-ToF-AMS data is a really valuable contribution to the field. The paper is very suitable for ACP, but unfortunately major (big major) revision are needed:

- Introduction. Decide if you want to focus on the study area, or on the techniques, decide one flow and report it. At the moment there is confusion.

- There are 144 references, really there is no need to add all these references, suggestion to cut to 60 max.

- Figure S8. Factors F1-F4 and F1-F6 in PMF analysis need to be better described and

[Figure]

named accordingly to the names of Factor 5 solution. Report also correlations among factors so the reader can understand how the factors evolve.

- The paper is very descriptive, and many papers are cited and referenced. There is no need. For example the whole section of Page 13 can be cut

- pg 14 delete all topic of aminoacid, it creates confusion. These markers used are not unique of aminoacids.

- naming. perhaps you want to simplify the naming, for example the aPOA may simply be anthropogenic organic aerosol (surely there will be a component that is secondary) and perhaps clearly stat that MOA POA and NOA are marine. NOA is marine, produced via secondary productions. Maybe start with "marine" or "anthopogenic" then "primary" or "secondary" then if it is Organic, nitrogen, MSA containing. Just a suggestion.

- Overall it is advised that the senior scientists co-authoring this paper suggest how to improve the flow of the current manuscript.

I congratulate to the authors (both corresponding authors in particular) for the impressive dataset collected - once the flow of this paper is improved, it will make a very important contribution in the field.

---

## Author Comment (AC1) · 21 Sep 2018

**Response to Referees' Comments:**

**Anonymous Referee #1**

(My ratings for both scientific and presentation quality are between "Good" and "Fair", so I give one "Good" and one "Fair".)

General Comments:

The authors performed 4 month-long field measurements across 53_N-53_S over the Atlantic Ocean from 2011-2012 and reported convincing source apportionment based on adequate and well-processed datasets obtained from HR-ToF-AMS and other techniques. Due to few number of similar studies that covered long time series and/or oceanic regions, the findings in this paper certainly provide valuable insights into the sources and origins of marine submicron atmospheric aerosols. Overall, the paper discussed relevant scientific questions within the scope of ACP journal with novel methods and datasets, and the results are generally (but not all) supportive to the interpretations and conclusions, in spite of some technical concerns and vague presentation or expression that need to be further supported, examined, or re-phrased. Therefore, I would recommend this paper be accepted for publication once the following specific comments are completely addressed.

Reply:

Thank you very much for your encouraging comments. We tried our best to improve the weak parts on both technical and language aspects. The detailed comments are responded point-to-point in the following text.

Suggestions for major improvements and revision:

1. The author(s) should have made best use of their valuable datasets as well as the

previously published studies, and emphasized the significance of their findings, if they also agree that they haven't done this enough in the abstract and the in the introduction. Besides, the author(s) should also, on one hand, carefully refer to previous studies that used similar techniques for marine aerosol, and on the other hand, include necessary comparisons (if available) in their own discussions. For example: The authors should add proper references to the sentences ended in: Page 10 Line 5; Page 11, Line 25; Page 12, Line 27; Page 12, Line 34; Page 13, Line 6; Page 16, Line 24 (Might be useful: Charlson et al., 1987, Nature; Bonsang et al. 1992, GRL; Yassaa et al., 2008, Env. Chem.; Shaw, Gantt, and Meskhidze, 2010, Advances in Meteorology).

Reply:

Thank you very much for your kind comments. According to referees' comments and co-authors' suggestion, we changed the title of this paper to "Organic aerosols over the Atlantic Ocean from 53°N to 53°S: a similar contribution of ocean and long-range transport". The abstract and introduction have been re-written to better stress 1) the key findings on Atlantic organic aerosol sources; 2) the motivation of this study; and 3) previous studies on the same topic or field.

We added the references to the sentences mentioned in examples, as well as in the not mentioned but necessary places. However, the number of references in the original version reached 144, which is too many as pointed out by the other referee. So we removed unnecessary references during re-organizing the whole manuscript. The detailed revisions to each mentioned sentence are showed below:

Page 10 Line 5: (Now in Page 8 Line 26) "Similar seasonal variation of the marine biogenic tracer MSA was also observed (Huang et al., 2017), suggesting the biogenic sources (i.e. phytoplankton) contributed significantly to sulfate (Charlson et al., 1987; Hoffmann et al., 2016)"

Page 11 Line 25: (Now in Page 10 Line 5) "This factor is well correlated with the marine tracer MSA ($R^2$ = 0.83, Figure 3), consequently linked to oxidation of DMS emitted by phytoplankton (Charlson et al., 1987; Gondwe et al., 2003)"

Page 12, Line 27: The original sentence "The diurnal cycle of MOA might have been weakened by averaging because the biological activities in autumn are usually lower than in spring." is subjective and not well supported by the chlorophyll a satellite map offered by the referee. We deleted the sentence and replaced it by the new one in Page 10 Line 32: "In order to focus on the atmospheric behavior of MOOA and exclude the influence from other chemical composition, a "MOOA dominating period" is selected for a case study (about 57 h from 19:40, 18.11.2012 to 04:20, 21.11.2012)."

Page 12, Line 34: (Now in Page11 Line 7) "Similar diurnal cycles are observed for MSA and sulfate, suggesting that MOOA, MSA and sulfate are formed via the same secondary pathway (Charlson et al., 1987; Gondwe et al., 2003; von Glasow and Crutzen, 2004)."

Page 13, Line 6: This sentence was deleted when re-organizing the whole paragraph.

Page 16, Line 24: The original sentences were: "In addition, the particles measured in the range from ~15°N to 15°S (close to the west and middle Africa) showed external mixing state by the HTDMA measurements (details in an accompany paper by Wu et al., in preparation). It is the typical property of BB emissions." Considering the mentioned paper will provide more detailed explanation and it is also arbitrary to define the "typical property of BB emissions", we deleted these two sentences in the end of paragraph.

2. Conclusions discussing causality or reasoning must be carefully examined. Just give a few examples: Page 9, Line 5: The authors attributed "insufficient offline samples" to the weak correlation between AMS and offline sea salt. Actually this might not be a reasonable explanation especially if they used AMS data collected from the exactly same periods of time during the offline filter sampling. The data size itself should not affect the R2, and the authors should also examine p-value of correlation for "meaningfulness". Furthermore, in this case, the authors should also clarify how they measured sea salts using the individual techniques and why they applied the method

from Ovadnevaite et al. (2012). For example, what ions were included as sea salts? Did they count Na+, Cl-, SO42-, K+, Mg2+, etc. in both? If NaCl accounted for different fraction from that in Ovadnevaite et al. (2012), was the scaling factor of 51 still suitable? Otherwise, the "therefore" in Line 6, did not explain why the same scaling factor was applied, considering the correlation and the coverage of time ("full year measurements in the reference") discussed above was not supportive, or not relevant.

Reply:

Thank you for your comments and suggestions. We reorganized the paragraph about sea salt estimation and add more technical details including:

1) We agree that the "insufficient offline samples" was not a reasonable explanation for the weak correlation. According to referee's suggestion, we performed the significance test (Spearman's correlation test because of non-normal distribution of the data) and the resulting p-value is 0.009, indicating the sea salt concentrations from two techniques (AMS and offline) are significantly correlated, that is, the correlation is meaningful. Now relative sentence is in Page 7 Line 32: "the p-value of the regression of AMS-derived sea salt with offline results is 0.009 (Spearman's correlation test), indicating that sea salt concentrations from AMS and offline methods are correlated significantly."

2) For the sea salt estimation based on $PM_1$ filter measurements, we use Na+ and $Cl^-$ ions to derive the sea salt concentration as applied in previous studies (Bates et al., 2001; Quinn et al., 2001): sea salt [$\mu$g m$^{-3}$] = $Cl^-$ [$\mu$g m$^{-3}$] + $Na^+$ [$\mu$g m$^{-3}$] x 1.47 ; where the factor of 1.47 is the seawater ratio of ($Na^+$ +$K^+$ +$Mg^{2+}$ + $Ca^{2+}$ +$SO_4^{2-}$ + $HCO_3^-$)/$Na^+$. So this estimation method can prevent the inclusion of non-sea-salt $K^+$, $Mg^{2+}$, $Ca^{2+}$, $SO_4^{2-}$ and $HCO_3^-$ in the sea salt mass, and also allow for the loss of chloride mass through chloride depletion processes (Bates et a., 2001). The estimation method was added to the caption of Figure S6.

3) For the sea salt estimation using HR-ToF-AMS, we use the method from Ovadnevaite et al. (2012). It is the first reported studies of sea salt estimation using AMS containing both laboratory calibrations using artificial sea salt and ambient

measurements, and the later studies (e.g. Schmale et al., 2013) followed the method (applied the similar scaling factor). The sea salts certainly include many ions once fragmentized by AMS, e.g. Na+, $Cl^-$, $SO_4^{2-}$, $K^+$, $Mg^{2+}$, $NaCl^+$, $Na_2Cl^+$ and so on (see details in Table 1 from Schmale et. al, 2013, shown below). However, the particulate sea salt mass concentration cannot be the simple sum of the ions on the list, because most of them could be contributed by non-sea-salt sources (e.g. $Cl^-$, $SO_4^{2-}$ and $K^+$ can be from continental transport) and some of the ions have too low intensity to be detected in ambient situation (e.g. metals and isotopes). Also, one of the main ions, $Na^+$, can vary significantly with the AMS vaporizer temperature (Ovadnevaite et al., 2012; Schmale et al., 2013). Therefore, Ovadnevaite et al. (2012) only used the $NaCl^+$ (m/z 57.95) as a surrogate of sea salt rather than the sea salt family ions, and we followed it for the same reason. In our study, the correlation slope between offline sea salt and AMS $NaCl^+$ ion was 62 ($\pm$ 6), not very far from the reference value, but with a mild correlation coefficient ($R^2 = 0.38$). We still prefer to use the scaling factor of 51 from the literature for several reasons: first, this factor could lead to better coherence (slope = 1.01) of estimated sea salt concentration between filter and AMS; second, most scattering dots (with factor of 51) are distributed within the uncertainty range derived from literature (Figure S6); and third, the factor of 51 is not too far from the slope derived in this study (62 $\pm$ 6) but can make our study consistent with the previous studies. Of course, the estimated sea salt mass concentration should be used and discussed with caution and we need to always be aware of its uncertainty.

According to the comments, we re-organized the paragraphs and removed the improper conjunction phrase "therefore". The revised sentence is now in Page 7 Line 28: "To be consistent with the literatures (Ovadnevaite et al., 2012; Ovadnevaite et al., 2014; Schmale et al., 2013), the scaling factor of 51 from the reference is applied to the sea salt surrogate ($NaCl^+$) to estimate the sea salt mass concentration in this study."

Table 1. Ion fragments considered within the sea salt family (Schmale et. al, 2013)

| Ion fragment | Exact mass | Ion fragment | Exact mass |
|---|---|---|---|
| Na | 22.99 | $Na_2^{35}Cl$ | 80.95 |
| Mg | 23.98 | $Na_2^{37}Cl$ | 82.94 |
| $^{25}Mg$ | 24.98 | $^{54}Fe^{35}Cl$ | 88.91 |
| $^{26}Mg$ | 25.98 | Zr | 89.90 |
| NaMg | 46.97 | $Fe^{35}Cl$ | 90.90 |
| Mn | 54.94 | $Fe^{37}Cl$ | 92.90 |
| OK | 54.96 | $^{94}Zr$ | 93.91 |
| Fe | 55.93 | $Mg^{35}Cl_2$ | 93.92 |
| Ni | 57.94 | $Mg^{37}Cl^{35}Cl$ | 95.92 |
| $Na^{35}Cl$ | 57.96 | $^{25}Mg^{37}Cl^{35}Cl$ | 96.92 |
| $Mg^{35}Cl$ | 58.95 | $Mg^{37}Cl_2$ | 97.92 |
| $Na^{37}Cl$ | 59.96 | $^{25}Mg^{37}Cl_2$ | 98.92 |
| $Mg^{37}Cl$ | 60.95 | $Fe^{35}Cl_2$ | 125.87 |
| $Cl_2$ | 69.94 | $Fe^{37}Cl^{35}Cl$ | 127.87 |
| $^{37}Cl^{35}Cl$ | 71.93 | | |

Page 10, Line 18: The author stated "The ammonium concentrations didn't follow a clear seasonal trend, although its precursor ammonia could be emitted from ocean (Ikeda, 2014; Johnson et al., 2008). The absence in seasonality suggests particulate ammonium during Polarstern cruises was contributed by both anthropogenic and biogenic sources." The deduction did not support their conclusion.

Reply:

Yes the causality of the sentences was not clear. We replaced them with the new ones: "The ammonium concentrations did not exhibit a seasonal difference between spring and autumn. The highest median value was found over the tropic Atlantic followed by the North Atlantic, while the lowest ones were in the South Hemisphere. Both continental pollutants via long-range transport and marine organism could be the origin of the ammonium or its precursor ammonia in MBL (Adams et al., 1999)." (Now in Page 9 Line 11).

Page 13, Line 29: "The diurnal variation of NOA shows clear peak in the afternoon, reaching the maximum while the global radiation starts decreasing (Figure 4), indicating that the NOA factor is certainly composed of secondary organic products."

The evidence is weak, and the authors should specify "global radiation" and cite papers that observed the similar diurnal trends of such a NOA factor, if available.

Reply:

We made major revision of the NOA part (now its name is changed to MNOA). Although the diurnal variation of NOA is very likely to be attributed to the secondary formation, it is hardly to find a NOA factors with similar diurnal pattern in previous studies. Sun et al., (2011) reported a NOA factor related to secondary transformation of amines from marine and industry sources but the factor showed a noon peak, different from our case. Due to the limited amount of AMS measurements in MBL, the comparison of specific marine factor become difficult so we can only make the speculation. New sentences started in Page 12 Line 3 "…and the diurnal variation of MNOA shows a broad afternoon peak with maximum at 1600UTC (Figure 4), similar to that of the amines-related secondary factor in the New York City despite maximum at noon time (Sun et al., 2011)." Besides, "global radiation" was defined in Page 9 Line 8 "… with global radiation (the sum of the direct solar radiation and diffuse radiation).". We have to use the global radiation because the solar radiation was not derived during the cruises.

Page 13, Line 20: The authors suggested "This can be useful for better estimation of marine DMS related SOA both in field measurements and in models". However, MSA as a fraction of SOA can vary largely and different from time to time (especially between summer and winter). In addition, MOA in this case might not be equivalent to SOA.

Reply:

We agree that MSA as a fraction of SOA may vary between different seasons, especially between summer and winter in which our measurements did not cover. Considering that AMS-PMF is an often-used method for distinguishing SOA and few other methods can provide more robust estimation of SOA, we think our result can at

least provide a hint on, not the whole marine SOA, but the SOA formed from DMS-oxidation in the measuring seasons, especially in spring. Of course, MSA cannot trace the portion of SOA which is formed from other pathways, e.g. secondary formation from gaseous amines. According to referee's comments and the explanation above, we revised the sentences in manuscript to be more cautious, emphasizing the season (spring) and SOA portion (DMS-related SOA) in/for which the MSA and scaling factor 1.79 can be applicable. The new sentence is now in Page 11 Lines 26: "We therefore infer that the relation between MSA and its concomitant (DMS-related) SOA is roughly stable over the Atlantic, and suggest to estimate MOOA mass concentration as the product of the MSA concentration multiplying the factor of 1.79." Further analysis of data from other oceans/seasons are needed in future to examine this correlation coefficient.

3. For better presentation quality and reading experience, the English language and scientific writing in this paper can be more precise and largely improved. Just give a few examples: Page 11, Line 9: "These S/C ratios derived from the PMF analysis tool contain however certain estimation uncertainties and have therefore to be used with caution." This seems to be a grammatically wrong sentence.

Reply:

Thank you very much for the comments. The authors of this paper have tried best to improve the language. We apologize that we did not make it be edited by a professional person/company due to very tight schedule of the author(s). We also noticed that a basic language correction and smoothing procedure would be provided by the journal as the last step of publication. Hope that would be helpful.

The sentence in Page 11, Line 9 is removed when re-organizing the text. This sentence is now changed to: "Note that the S/C ratios derived from the PMF analysis tool have to be used with caution because of calculation uncertainties (Aiken et al., 2007), but they can still provide indication on significance of sulfur when calculated with the same

tool among the factors from the same dataset." (Page 10 Line 7)

Page 12, Line 24: "The minimum of the diurnal variation (0.04 μg m$^{-3}$) appears around 09:00, probably linking to the increase of mixing layer in morning." This sentence needs to be re-phrased and also supported with references.

Reply:

Based on the re-analysis of the MOOA diurnal variation, we think it is insufficient to attribute the minimum to the dilution effect of the rising boundary layer. Because the drop of the MOOA concentration was not found at the similar time point during the MOOA-dominating period. So, this sentence is removed.

Page 10, Line 27: I think it is more precise to say "57 hours" rather than "about 2 consecutive days", unless there was an interruption.

Reply:

Done.

Page 13, Line 6: "the this OA component". Despite the grammatical error and lack of references, "OA component" was vague in the context.

Reply:

Thanks for reminding this. We reorganized this paragraph and deleted the mentioned sentence.

Page 17, Line 17: In this paragraph, the author said "still questionable" and then "This suggests… could be not correlated". This led to confusion due to the inappropriate English or logical expression.

Reply:

We apologize for the unclear causality here. We improved the sentences as: "This does not conflict with the speculation that the MHOA is related with marine primary

emissions, because the mass fraction of organics in the sea spray aerosol was found to be size-dependent: increasing with the decreasing particle size (Gantt et al., 2011; Quinn et al., 2015). The enrichment factor of organic compounds, i.e. the ratio between organic carbon in sea spray aerosol and that in sea water, is also largely influenced by the particle size (Quinn et al., 2015). In addition, the transfer of organic matters from seawater to the particles is chemoselevtive, more complicated than the inorganic sea salt (Schmitt-Kopplin et al., 2012)".

Other technical and specific comments to be addressed:

1. Generally when discussing seasonality, the difference between "spring & autumn" might not be as distinct as that between "summer & winter", in term of many factors such as meteorological parameters and marine bioactivity. Besides the "spring vs. autumn" comparison, the authors may also want to look into "spring/autumn vs. tropic". In addition, their measurements on board was changing with time and location at the same time, so this will be different from those studies took place at a ground site over seasons. I wonder if the authors would like to make some comments on these.

Reply:

Thanks for the comments. Yes, the meteorological parameters such as temperatures and RH were not very different between spring and autumn as between summer and winter. We added the description on tropical case and compared the species mass concentrations in spring, autumn and tropic (Session 3.1.2). Although no big difference was found between spring and autumn for organics, sea salt, nitrate and so on, sulfate showed very discrepant average or median mass concentration in spring and autumn, maybe related to different biological activities. This may suggest even with the similar temperatures and RH, the seasonal events such as biological activities may still influence the aerosol chemical composition.

Considering the comparability between the mobile platform and stationary site, we admit there could be big difference caused by marine biomass distribution, e.g. more

dense phytoplankton group near the coastal region than the remote ocean. But because of very limited amount of the mobile measurements over the ocean, it is quite difficult to find records for the similar regions. So we collected the aerosol chemical composition in several regions covering the ship tracks, and checked if they were comparable to our results. In future it would be helpful to have more information of submicron aerosols over the ocean based on satellite data.

2. The authors should try to clarify the influences from the "open oceans", "marine", and "coastal" when interpreting results in the discussions, even though the boundaries might be blurry. For example, on Page 11, Line 30, the author stated "The S/C ratio of the MOA factor is also over twice that of marine factor observed in Paris (0.013, Crippa et al., 2013b), implying a stronger influence from marine phytoplankton on aerosol particles over the ocean than those in the coast city.", but actually the abundance of phytoplankton can be much higher in the coastal areas. See https://earthobservatory.nasa.gov/GlobalMaps/view.php?d1=MY1DMM_CHLORA

Reply:

Thanks for the comments and website link. Due to the limited on-board measurements, we did not find the S/C ratios from offshore sites or open oceans in previous studies. So the S/C ratio from coastal measurements in Paris was used as a reference for comparison. We noticed that the comparison is not sufficient to support the conclusion of "stronger influence … over the ocean than those in the coast city", and the S/C ratio should be used with caution because of calculation uncertainties. So this conclusion is removed and we only compare the S/C ratios among the OA factors in this study. Now in Page 10, Line 6: "This leads to a high S/C ratio (0.030), which is 10 to 30 times higher than that of other factors (Figure 3)." Nevertheless, it is important to stress that organosulfates with biogenic sources may not correlate with chlorophyll a level, as the former is produced from the secondary pathway and the latter is the indicator of the primary biogenic mass (Huang et al., 2017).

3. The authors are suggested to add discussions for organosulfates, since they can make a considerable contribution to continental SOA masses at certain locations, and also derived from the same biogenic precursors over the oceans. For example, how is this class of compounds measured using AMS? Was it included in organics or sulfate, or neither?

Reply:

Thanks a lot for the suggestion. Organosulfates (except MSA) are also one of the important components of the marine SOA transformed from the precursors such as isoprene, monoterpenes and so on (Claeys et al., 2010; Surratt et al., 2007). We added small discussion on the organosulfates in the revised manuscript to stress the existence of organosulfates (Page 10 Line 28): it is well-known that isoprene and monoterpenes oxidation lead also to the formation of organosulfate compounds (Claeys et al., 2010; Fu et al., 2011; Iinuma et al., 2007; Surratt et al., 2008; Surratt et al., 2007), which can contribute to the $C_xS_y^+$ fragments observed in the MOOA factor." We hesitated to discuss more about the organosulfates in this paper because a parallel paper focusing on the organosulfates is in preparation, which included detailed analysis on a sub-dataset of Polarstern measurements. The paragraph below may answer referee's questions:

Our MOA (now changed to MOOA) mainly includes MSA fragments (as shown in Figure 1), while the contribution of organosulfates to MOOA may be tiny as found in previous marine study (Claeys et al., 2010). Using AMS, the fragments of organosulfate (e.g. MSA) are recognized as sulfate and organics (Figure 1). The quantification of organosulfates requires the laboratory calibrations using standard chemicals of known organosulfates (Huang et al., 2015; Huang et al., 2017).

[Figure]

Figure 1 Mass spectra of MOOA factor (CH and CS ions) and pure MSA (CH, CS and SO ions)

4. Last but not least, the authors should revise the manuscript carefully by their own. Just give a few examples: 1) Page 12, Line 34: "Figure 4" – should this be Figure 5?

Reply:

Thanks for the detailed comments! In original sentence "Figure 4" was mentioned for "the average case". In order to be more precise, the sentence is changed to " The diurnal pattern for this specific period (Figure 6b), with minimum of 0.11 µg m$^{-3}$ at 07:00 and maximum of 0.25 µg m$^{-3}$ at 16:00, is more noticeable than the average case (Figure 4b)." (Page 11 Line 6).

2) Acronym: define before use. For example, "SOA" was not defined but used in the abstract; "OA" was firstly defined on Page 10, Line 30 in the main text; "biomass burning" was defined but not used in many places.

Reply:

We checked the manuscript and corrected the use of acronyms. The definition of acronyms was added to the abstract: Page 1 Line 17 for OA, Page 2 Line 2 for SOA. Biomass burning (BB) was defined in Page 12 Line 22 and the abbreviation is mainly

used in the section 3.2.5 Combustion oxygenated organic aerosol (Comb-OOA), e.g. Page 15 Line 2, "the average BB organic aerosols", Line 15 " some often used BB tracers".

3) Please be consistent when using terms such as "fPeak" or "fpeak", "CxHyO" or "CxHyO1". 4) Please be consistent about adding a "_" between numerical values and their units. 5) Please specify "CxSj+" on Page 12, Line 21.

Reply:

We went through the text and uniform the terms: e.g. fPeak, $C_xH_yO^+$ and $C_xS_y^+$. We also uniform the format of values and units: put a blank between them.

**References**

Aiken, A. C., DeCarlo, P. F., & Jimenez, J. L. (2007). Elemental Analysis of Organic Species with Electron Ionization High-Resolution Mass Spectrometry. *Anal. Chem., 79*(21), 8350-8358.

Bates, T. S., Quinn, P. K., Coffman, D. J., Johnson, J. E., Miller, T. L., et al. (2001). Regional physical and chemical properties of the marine boundary layer aerosol across the Atlantic during Aerosols99: An overview. *J. Geophys. Res. - Atmos., 106*(D18), 20767-20782.

Charlson, R. J., Lovelock, J. E., Andreae, M. O., & Warren, S. G. (1987). Oceanic phytoplankton, atmospheric sulphur, cloud albedo and climate. [10.1038/326655a0]. *Nature, 326*(6114), 655-661.

Claeys, M., Wang, W., Vermeylen, R., Kourtchev, I., Chi, X. G., et al. (2010). Chemical characterisation of marine aerosol at Amsterdam Island during the austral summer of 2006-2007. *J. Aerosol Sci., 41*(1), 13-22.

Fu, P., Kawamura, K., & Miura, K. (2011). Molecular characterization of marine organic aerosols collected during a round-the-world cruise. *J. Geophys. Res. - Atmos., 116*(D13), D13302.

Gantt, B., Meskhidze, N., Facchini, M. C., Rinaldi, M., Ceburnis, D., et al. (2011). Wind speed dependent size-resolved parameterization for the organic mass fraction of sea spray aerosol. *Atmos. Chem. Phys., 11*(16), 8777-8790.

Gondwe, M., Krol, M., Gieskes, W., Klaassen, W., & de Baar, H. (2003). The contribution of ocean-leaving DMS to the global atmospheric burdens of DMS, MSA, SO2, and NSS SO4=. *Global Biogeochem. Cy., 17*(2), 1056.

Hoffmann, E. H., Tilgner, A., Schrödner, R., Bräuer, P., Wolke, R., et al. (2016). An advanced modeling study on the impacts and atmospheric implications of multiphase dimethyl sulfide chemistry. *Proceedings of the National Academy of Sciences, 113*(42), 11776-11781.

Huang, D. D., Li, Y. J., Lee, B. P., & Chan, C. K. (2015). Analysis of Organic Sulfur Compounds in Atmospheric Aerosols at the HKUST Supersite in Hong Kong Using HR-ToF-AMS. *Environ. Sci. Technol., 49*(6), 3672-3679.

Huang, S., Poulain, L., van Pinxteren, D., van Pinxteren, M., Wu, Z., et al. (2017). Latitudinal and Seasonal Distribution of Particulate MSA over the Atlantic using a Validated Quantification Method with HR-ToF-AMS. *Environ. Sci. Technol., 51*(1), 418-426.

Iinuma, Y., Müller, C., Böge, O., Gnauk, T., & Herrmann, H. (2007). The formation of organic sulfate esters in the limonene ozonolysis secondary organic aerosol (SOA) under acidic conditions. *Atmos. Environ., 41*(27), 5571-5583.

Ovadnevaite, J., Ceburnis, D., Canagaratna, M., Berresheim, H., Bialek, J., et al. (2012). On the effect of wind speed on submicron sea salt mass concentrations and source fluxes. *J. Geophys. Res., 117*(D16), D16201.

Quinn, P. K., Coffman, D. J., Bates, T. S., Miller, T. L., Johnson, J. E., et al. (2001). Dominant aerosol chemical components and their contribution to extinction during the Aerosols99 cruise across the Atlantic. *J. Geophys. Res. - Atmos., 106*(D18), 20783-20809.

Quinn, P. K., Collins, D. B., Grassian, V. H., Prather, K. A., & Bates, T. S. (2015). Chemistry and Related Properties of Freshly Emitted Sea Spray Aerosol. *Chem. Rev.*

Schmale, J., Schneider, J., Nemitz, E., Tang, Y. S., Dragosits, U., et al. (2013). Sub-Antarctic marine aerosol: dominant contributions from biogenic sources. *Atmos. Chem. Phys., 13*(17), 8669-8694.

Schmitt-Kopplin, P., Liger-Belair, G., Koch, B. P., Flerus, R., Kattner, G., et al. (2012). Dissolved organic matter in sea spray: a transfer study from marine surface water to aerosols. *Biogeosciences, 9*(4), 1571-1582.

Sun, Y. L., Zhang, Q., Schwab, J. J., Demerjian, K. L., Chen, W. N., et al. (2011). Characterization of the sources and processes of organic and inorganic aerosols in New York city with a high-resolution time-of-flight aerosol mass apectrometer. *Atmos. Chem. Phys., 11*(4), 1581-1602.

Surratt, J. D., Gómez-González, Y., Chan, A. W. H., Vermeylen, R., Shahgholi, M., et al. (2008). Organosulfate Formation in Biogenic Secondary Organic Aerosol. *The Journal of Physical Chemistry A, 112*(36), 8345-8378.

Surratt, J. D., Kroll, J. H., Kleindienst, T. E., Edney, E. O., Claeys, M., et al. (2007). Evidence for Organosulfates in Secondary Organic Aerosol. *Environ. Sci. Technol., 41*(2), 517-527.

---

## Author Response (AR1)

**Response to Referees' Comments:**

**Anonymous Referee #1**

(My ratings for both scientific and presentation quality are between "Good" and "Fair", so I give one "Good" and one "Fair".)

General Comments:

The authors performed 4 month-long field measurements across 53_N-53_S over the Atlantic Ocean from 2011-2012 and reported convincing source apportionment based on adequate and well-processed datasets obtained from HR-ToF-AMS and other techniques. Due to few number of similar studies that covered long time series and/or oceanic regions, the findings in this paper certainly provide valuable insights into the sources and origins of marine submicron atmospheric aerosols. Overall, the paper discussed relevant scientific questions within the scope of ACP journal with novel methods and datasets, and the results are generally (but not all) supportive to the interpretations and conclusions, in spite of some technical concerns and vague presentation or expression that need to be further supported, examined, or re-phrased. Therefore, I would recommend this paper be accepted for publication once the following specific comments are completely addressed.

Reply:

Thank you very much for your encouraging comments. We tried our best to improve the weak parts on both technical and language aspects. The detailed comments are responded point-to-point in the following text.

Suggestions for major improvements and revision:

1. The author(s) should have made best use of their valuable datasets as well as the

previously published studies, and emphasized the significance of their findings, if they also agree that they haven't done this enough in the abstract and the in the introduction. Besides, the author(s) should also, on one hand, carefully refer to previous studies that used similar techniques for marine aerosol, and on the other hand, include necessary comparisons (if available) in their own discussions. For example: The authors should add proper references to the sentences ended in: Page 10 Line 5; Page 11, Line 25; Page 12, Line 27; Page 12, Line 34; Page 13, Line 6; Page 16, Line 24 (Might be useful: Charlson et al., 1987, Nature; Bonsang et al. 1992, GRL; Yassaa et al., 2008, Env. Chem.; Shaw, Gantt, and Meskhidze, 2010, Advances in Meteorology).

Reply:

Thank you very much for your kind comments. According to referees' comments and co-authors' suggestion, we changed the title of this paper to "Organic aerosols over the Atlantic Ocean from 53°N to 53°S: similar contributions from ocean and long-range transport". The abstract and introduction have been re-written to better stress 1) the key findings on Atlantic organic aerosol sources; 2) the motivation of this study; and 3) previous studies on the same topic or field.

We added the references to the sentences mentioned in examples, as well as in the not mentioned but necessary places. However, the number of references in the original version reached 144, which is too many as pointed out by the other referee. So we removed unnecessary references during re-organizing the whole manuscript (shrinking to 87 references). The detailed revisions to each mentioned sentence are showed below:

Page 10 Line 5: (Now in Page 8 Line 24) "A similar seasonal variation of the marine biogenic tracer MSA was also observed (Huang et al., 2017), suggesting the biogenic sources (i.e. phytoplankton) contributed significantly to sulfate (Charlson et al., 1987; Hoffmann et al., 2016)"

Page 11 Line 25: (Now in Page 10 Line 4) "This factor is well correlated with the marine tracer MSA ($R^2 = 0.83$, Figure 3) and is consequently linked to the oxidation of DMS emitted by phytoplankton (Charlson et al., 1987; Gondwe et al., 2003)."

Page 12, Line 27: The original sentence "The diurnal cycle of MOA might have been weakened by averaging because the biological activities in autumn are usually lower than in spring." is subjective and not well supported by the chlorophyll a satellite map offered by the referee. We deleted the sentence and replaced it by the new one in Page 10 Line 32: "To focus on the atmospheric behavior of MOOA and exclude the influence from other chemical composition, a "MOOA dominating period" is selected for a case study (about 57 h from 19:40, 18.11.2012 to 04:20, 21.11.2012)."

Page 12, Line 34: (Now in Page11 Line 7) "Similar diurnal cycles are observed for MSA and sulfate, suggesting that MOOA, MSA and sulfate are formed via the same secondary pathway (Charlson et al., 1987; Gondwe et al., 2003; von Glasow and Crutzen, 2004)."

Page 13, Line 6: This sentence was deleted when re-organizing the whole paragraph.

Page 16, Line 24: The original sentences were: "In addition, the particles measured in the range from ~15°N to 15°S (close to the west and middle Africa) showed external mixing state by the HTDMA measurements (details in an accompany paper by Wu et al., in preparation). It is the typical property of BB emissions." Considering the mentioned paper will provide more detailed explanation and it is also arbitrary to define the "typical property of BB emissions", we deleted these two sentences in the end of paragraph.

2. Conclusions discussing causality or reasoning must be carefully examined. Just give a few examples: Page 9, Line 5: The authors attributed "insufficient offline samples" to the weak correlation between AMS and offline sea salt. Actually this might not be a reasonable explanation especially if they used AMS data collected from the exactly same periods of time during the offline filter sampling. The data size itself should not affect the $R^2$, and the authors should also examine p-value of correlation for "meaningfulness". Furthermore, in this case, the authors should also clarify how they measured sea salts using the individual techniques and why they applied the method

from Ovadnevaite et al. (2012). For example, what ions were included as sea salts? Did they count Na+, Cl-, SO42-, K+, Mg2+, etc. in both? If NaCl accounted for different fraction from that in Ovadnevaite et al. (2012), was the scaling factor of 51 still suitable? Otherwise, the "therefore" in Line 6, did not explain why the same scaling factor was applied, considering the correlation and the coverage of time ("full year measurements in the reference") discussed above was not supportive, or not relevant.

Reply:

Thank you for your comments and suggestions. We reorganized the paragraph about sea salt estimation and added more technical details including:

1) We agree that the "insufficient offline samples" was not a reasonable explanation for the weak correlation. According to referee's suggestion, we performed the significance test (Spearman's correlation test because of non-normal distribution of the data) and the resulting p-value is 0.009, indicating the sea salt concentrations from two techniques (AMS and offline) are significantly correlated, that is, the correlation is meaningful. Now relative sentence is in Page 7 Line 29: "the p-value of the regression of AMS-derived sea salt with offline results is 0.009 (Spearman's correlation test), indicating that sea salt concentrations from AMS and offline methods are correlated significantly."

2) For the sea salt estimation based on $PM_1$ filter measurements, we use $Na^+$ and $Cl^-$ ions to derive the sea salt concentration as applied in previous studies (Bates et al., 2001; Quinn et al., 2001): sea salt $[\mu g\ m^{-3}] = Cl^-\ [\mu g\ m^{-3}] + Na^+\ [\mu g\ m^{-3}]$ x 1.47 ; where the factor of 1.47 is the seawater ratio of $(Na^+ + K^+ + Mg^{2+} + Ca^{2+} + SO_4^{2-} + HCO_3^-)/Na^+$. So this estimation method can prevent the inclusion of non-sea-salt $K^+$, $Mg^{2+}$, $Ca^{2+}$, $SO_4^{2-}$ and $HCO_3^-$ in the sea salt mass, and also allow for the loss of chloride mass through chloride depletion processes (Bates et a., 2001). The estimation method was added to the caption of Figure S6.

3) For the sea salt estimation using HR-ToF-AMS, we use the method from Ovadnevaite et al. (2012). It is the first reported studies of sea salt estimation using AMS containing both laboratory calibrations using artificial sea salt and ambient

measurements, and the later studies (e.g. Schmale et al., 2013) followed the method (applied the similar scaling factor). The sea salts certainly include many ions once fragmentized by AMS, e.g. Na+, $Cl^-$, $SO_4^{2-}$, $K^+$, $Mg^{2+}$, $NaCl^+$, $Na_2Cl^+$ and so on (see details in Table 1 from Schmale et. al, 2013, shown below). However, the particulate sea salt mass concentration cannot be the simple sum of the ions on the list, because most of them could be contributed by non-sea-salt sources (e.g. $Cl^-$, $SO_4^{2-}$ and $K^+$ can be from continental transport) and some of the ions have too low intensity to be detected in ambient situation (e.g. metals and isotopes). Also, one of the main ions, $Na^+$, can vary significantly with the AMS vaporizer temperature (Ovadnevaite et al., 2012; Schmale et al., 2013). Therefore, Ovadnevaite et al. (2012) only used the $NaCl^+$ (m/z 57.95) as a surrogate of sea salt rather than the sea salt family ions, and we followed it for the same reason. In our study, the correlation slope between offline sea salt and AMS $NaCl^+$ ion was 62 ($\pm$ 6), not very far from the reference value, but with a mild correlation coefficient ($R^2$ = 0.38). We still prefer to use the scaling factor of 51 from the literature for several reasons: first, this factor could lead to better coherence (slope = 1.01) of estimated sea salt concentration between filter and AMS; second, most scattering dots (with factor of 51) are distributed within the uncertainty range derived from literature (Figure S6); and third, the factor of 51 is not too far from the slope derived in this study (62 $\pm$ 6) but can make our study consistent with the previous studies. Of course, the estimated sea salt mass concentration should be used and discussed with caution and we need to always be aware of its uncertainty.

According to the comments, we re-organized the paragraphs and removed the improper conjunction phrase "therefore". The revised sentence is now in Page 7 Line 25: "To be consistent with the literature (Ovadnevaite et al., 2012; Ovadnevaite et al., 2014; Schmale et al., 2013), the scaling factor of 51 from the reference is applied to the sea salt surrogate ($NaCl^+$) to estimate the sea salt mass concentration in this study."

Table 1. Ion fragments considered within the sea salt family (Schmale et. al, 2013)

| Ion fragment | Exact mass | Ion fragment | Exact mass |
|---|---|---|---|
| Na | 22.99 | $Na_2^{35}Cl$ | 80.95 |
| Mg | 23.98 | $Na_2^{37}Cl$ | 82.94 |
| $^{25}Mg$ | 24.98 | $^{54}Fe^{35}Cl$ | 88.91 |
| $^{26}Mg$ | 25.98 | Zr | 89.90 |
| NaMg | 46.97 | $Fe^{35}Cl$ | 90.90 |
| Mn | 54.94 | $Fe^{37}Cl$ | 92.90 |
| OK | 54.96 | $^{94}Zr$ | 93.91 |
| Fe | 55.93 | $Mg^{35}Cl_2$ | 93.92 |
| Ni | 57.94 | $Mg^{37}Cl^{35}Cl$ | 95.92 |
| $Na^{35}Cl$ | 57.96 | $^{25}Mg^{37}Cl^{35}Cl$ | 96.92 |
| $Mg^{35}Cl$ | 58.95 | $Mg^{37}Cl_2$ | 97.92 |
| $Na^{37}Cl$ | 59.96 | $^{25}Mg^{37}Cl_2$ | 98.92 |
| $Mg^{37}Cl$ | 60.95 | $Fe^{35}Cl_2$ | 125.87 |
| $Cl_2$ | 69.94 | $Fe^{37}Cl^{35}Cl$ | 127.87 |
| $^{37}Cl^{35}Cl$ | 71.93 | | |

Page 10, Line 18: The author stated "The ammonium concentrations didn't follow a clear seasonal trend, although its precursor ammonia could be emitted from ocean (Ikeda, 2014; Johnson et al., 2008). The absence in seasonality suggests particulate ammonium during Polarstern cruises was contributed by both anthropogenic and biogenic sources." The deduction did not support their conclusion.

Reply:

Yes the causality of the sentences was not clear. We replaced them with the new ones: "The ammonium concentrations did not exhibit a seasonal difference between spring and autumn. The highest median value was found over the tropic Atlantic, followed by the North Atlantic, while the lowest median value was in the South Hemisphere. Both continental emissions via long-range transport and marine organism could be the origin of the ammonium or its precursor ammonia in the MBL (Adams et al., 1999)." (Now in Page 9 Line 9).

Page 13, Line 29: "The diurnal variation of NOA shows clear peak in the afternoon,

reaching the maximum while the global radiation starts decreasing (Figure 4), indicating that the NOA factor is certainly composed of secondary organic products." The evidence is weak, and the authors should specify "global radiation" and cite papers that observed the similar diurnal trends of such a NOA factor, if available.

Reply:

We made major revision of the NOA part (now its name is changed to MNOA). Although the diurnal variation of NOA is very likely to be attributed to the secondary formation, it is hardly to find a NOA factors with similar diurnal pattern in previous studies. Sun et al., (2011) reported a NOA factor related to secondary transformation of amines from marine and industry sources but the factor showed a noon peak, comparable to our case (afternoon peak around 16:00UTC). Due to the limited amount of AMS measurements in the MBL, the comparison of specific marine factor become difficult so we can only make the speculation. New sentences started in Page 12 Line 3 "…and the diurnal variation of MNOA shows a broad afternoon peak with maximum at 16:00 UTC (Figure 4), similar to that of the amines-related secondary factor in the New York City showing a diurnal pattern with maximum at noon time (Sun et al., 2011)." Besides, "global radiation" was defined in Page 9 Line 29 "… with global radiation (the sum of the direct solar radiation and diffuse radiation)". We have to use the global radiation because the solar radiation data was not available during the cruises.

Page 13, Line 20: The authors suggested "This can be useful for better estimation of marine DMS related SOA both in field measurements and in models". However, MSA as a fraction of SOA can vary largely and different from time to time (especially between summer and winter). In addition, MOA in this case might not be equivalent to SOA.

Reply:

We agree that MSA as a fraction of SOA may vary between different seasons, especially between summer and winter in which our measurements did not cover.

Considering that AMS-PMF is an often-used method for distinguishing SOA and few other methods can provide more robust estimation of SOA, we think our result can at least provide a hint on, not the whole marine SOA, but the SOA formed from DMS-oxidation in the measuring seasons, especially in spring. Of course, MSA cannot trace the portion of SOA which is formed from other pathways, e.g. secondary formation from gaseous amines. According to referee's comments and the explanation above, we revised the sentences in manuscript to be more cautious, emphasizing the season (spring) and SOA portion (DMS-related SOA) in/for which the MSA and scaling factor 1.79 can be applicable. The new sentence is now in Page 11 Lines 26: "We therefore infer that the relation between MSA and its concomitant (DMS-related) SOA is roughly stable over the Atlantic, and suggest estimating MOOA mass concentration as the product of the MSA concentration multiplied by a factor of 1.79, which may be useful for a better estimation of marine DMS-related SOA both in field measurements and in models." Further analysis of data from other oceans/seasons are needed in future to examine this correlation coefficient.

3. For better presentation quality and reading experience, the English language and scientific writing in this paper can be more precise and largely improved. Just give a few examples: Page 11, Line 9: "These S/C ratios derived from the PMF analysis tool contain however certain estimation uncertainties and have therefore to be used with caution." This seems to be a grammatically wrong sentence.

Reply:

Thank you very much for the comments. The authors of this paper have tried best to improve the language. During the period after authors' reply, we finally had the manuscript improved by a professional company (American Journal Expert) for a standard editing. Now the words and grammar mistakes are already eliminated.

The sentence in Page 11, Line 9 is removed when re-organizing the text. This sentence is now changed to: "Note that the S/C ratios derived from the PMF analysis tool have

to be used with caution because of calculation uncertainties (Aiken et al., 2007), but they can still provide an indication of the significance of sulfur when calculated with the same tool among the factors from the same dataset." (Page 10 Line 9)

Page 12, Line 24: "The minimum of the diurnal variation (0.04 μg m$^{-3}$) appears around 09:00, probably linking to the increase of mixing layer in morning." This sentence needs to be re-phrased and also supported with references.

Reply:

Based on the re-analysis of the MOOA diurnal variation, we think it is insufficient to attribute the minimum to the dilution effect of the rising boundary layer. Because the drop of the MOOA concentration was not found at the similar time point during the MOOA-dominating period. So, this sentence is removed.

Page 10, Line 27: I think it is more precise to say "57 hours" rather than "about 2 consecutive days", unless there was an interruption.

Reply:

Thanks for the suggestion. This has been done.

Page 13, Line 6: "the this OA component". Despite the grammatical error and lack of references, "OA component" was vague in the context.

Reply:

Thanks for reminding this. We reorganized this paragraph and deleted the mentioned sentence.

Page 17, Line 17: In this paragraph, the author said "still questionable" and then "This suggests… could be not correlated". This led to confusion due to the inappropriate English or logical expression.

Reply:

We apologize for the unclear causality here. We improved the sentences as (now in Page 13 Line 21): "This trend does not conflict with the speculation that MHOA is related to marine primary emissions because the mass fraction of organics in the sea spray aerosol was found to be size-dependent: increasing with decreasing particle size (Gantt et al., 2011; Quinn et al., 2015). The enrichment factor of organic compounds, i.e., the ratio between organic carbon in sea spray aerosols and that in sea water, is also largely influenced by particle size (Quinn et al., 2015). In addition, the transfer of organic matter from seawater to the particles is chemoselective and more complicated 25 than it is for inorganic sea salt ( (Schmitt-Kopplin et al., 2012)".

Other technical and specific comments to be addressed:

1. Generally when discussing seasonality, the difference between "spring & autumn" might not be as distinct as that between "summer & winter", in term of many factors such as meteorological parameters and marine bioactivity. Besides the "spring vs. autumn" comparison, the authors may also want to look into "spring/autumn vs. tropic". In addition, their measurements on board was changing with time and location at the same time, so this will be different from those studies took place at a ground site over seasons. I wonder if the authors would like to make some comments on these.

Reply:

Thanks for the comments. Yes, the meteorological parameters such as temperatures and RH were not very different between spring and autumn as between summer and winter. We added the description on tropical case and compared the species mass concentrations in spring, autumn and tropic (Session 3.1.2). Although no big difference was found between spring and autumn for organics, sea salt, nitrate and so on, sulfate showed very discrepant average or median mass concentration in spring and autumn, maybe related to different biological activities. This may suggest even with the similar temperatures and RH, the seasonal events such as biological activities may still influence the aerosol chemical composition.

Considering the comparability between the mobile platform and stationary site, we admit there could be big difference caused by marine biomass distribution, e.g. more dense phytoplankton group near the coastal region than the remote ocean. But because of very limited amount of the mobile measurements over the ocean, it is quite difficult to find records for the similar regions. So we collected the aerosol chemical composition in several regions covering the ship tracks, and checked if they were comparable to our results. In future it would be helpful to have more information of submicron aerosols over the ocean based on satellite data.

2. The authors should try to clarify the influences from the "open oceans", "marine", and "coastal" when interpreting results in the discussions, even though the boundaries might be blurry. For example, on Page 11, Line 30, the author stated "The S/C ratio of the MOA factor is also over twice that of marine factor observed in Paris (0.013, Crippa et al., 2013b), implying a stronger influence from marine phytoplankton on aerosol particles over the ocean than those in the coast city.", but actually the abundance of phytoplankton can be much higher in the coastal areas. See https://earthobservatory.nasa.gov/GlobalMaps/view.php?d1=MY1DMM_CHLORA

Reply:

Thanks for the comments and website link. Due to the limited on-board measurements, we did not find the S/C ratios from offshore sites or open oceans in previous studies. So the S/C ratio from coastal measurements in Paris was used as a reference for comparison. We noticed that the comparison is not sufficient to support the conclusion of "stronger influence … over the ocean than those in the coast city", and the S/C ratio should be used with caution because of calculation uncertainties. So this conclusion is removed and we only compare the S/C ratios among the OA factors in this study. Now in Page 10, Line 8: "This leads to a high S/C ratio (0.030), which is 10 to 30 times higher than that of other factors (Figure 3)." Nevertheless, it is important to stress that organosulfates with biogenic sources may not correlate with chlorophyll a level, as the

former is produced from the secondary pathway and the latter is the indicator of the primary biogenic mass (Huang et al., 2017).

3. The authors are suggested to add discussions for organosulfates, since they can make a considerable contribution to continental SOA masses at certain locations, and also derived from the same biogenic precursors over the oceans. For example, how is this class of compounds measured using AMS? Was it included in organics or sulfate, or neither?

Reply:

Thanks a lot for the suggestion. Organosulfates (except MSA) are also one of the important components of the marine SOA transformed from the precursors such as isoprene, monoterpenes and so on (Claeys et al., 2010; Surratt et al., 2007). We added small discussion on the organosulfates in the revised manuscript to stress the existence of organosulfates (Page 10 Line 28): It is well known that isoprene and monoterpene oxidation also leads to the formation of organosulfate compounds (Claeys et al., 2010; Fu et al., 2011; Iinuma et al., 2007; Surratt et al., 2008; Surratt et al., 2007), which can contribute to the $C_xS_y^+$ fragments observed in the MOOA factor." We hesitated to discuss more about the organosulfates in this paper because a parallel paper focusing on the organosulfates is in preparation, which included detailed analysis on a sub-dataset of Polarstern measurements. The paragraph below may answer referee's questions:

Our MOA (now changed to MOOA) mainly includes MSA fragments (as shown in Figure 1), while the contribution of organosulfates to MOOA may be tiny as found in previous marine study (Claeys et al., 2010). Using AMS, the fragments of organosulfate (e.g. MSA) are recognized as sulfate and organics (Figure 1). The quantification of organosulfates requires the laboratory calibrations using standard chemicals of known organosulfates (Huang et al., 2015; Huang et al., 2017).

[Figure]

Figure 1 Mass spectra of MOOA factor (CH and CS ions) and pure MSA (CH, CS and SO ions)

4. Last but not least, the authors should revise the manuscript carefully by their own. Just give a few examples: 1) Page 12, Line 34: "Figure 4" – should this be Figure 5?

Reply:

Thanks for the detailed comments! In original sentence "Figure 4" was mentioned for "the average case". In order to be more precise, the sentence is changed to " The diurnal pattern for this specific period (Figure 6b), with a minimum of 0.11 µg m$^{-3}$ (MOOA mass concentration) at 07:00 and a maximum of 0.25 µg m$^{-3}$ at 16:00, was more noticeable than the average case (Figure 4b)." (Page 11 Line 6).

2) Acronym: define before use. For example, "SOA" was not defined but used in the abstract; "OA" was firstly defined on Page 10, Line 30 in the main text; "biomass burning" was defined but not used in many places.

Reply:

We checked the manuscript and corrected the use of acronyms. The definition of acronyms was added to the abstract: Page 1 Line 16 for OA, Page 2 Line 1 for SOA. Biomass burning (BB) was defined in Page 12 Line 21 and the abbreviation is mainly

used in the section 3.2.5 Combustion oxygenated organic aerosol (Comb-OOA), e.g. Page 15 Line 2, "…the average BB organic aerosols…", Line 15 "The absence of these BB tracers…".

3) Please be consistent when using terms such as "fPeak" or "fpeak", "CxHyO" or "CxHyO1". 4) Please be consistent about adding a "_" between numerical values and their units. 5) Please specify "CxSj+" on Page 12, Line 21.

Reply:

We went through the text and uniform the terms: e.g. fPeak, $C_xH_yO^+$ and $C_xS_y^+$. We also uniform the format of values and units: put a blank between them.

**Response to Referees' Comments:**

**Anonymous Referee #2**

The technical aspect of the current paper is very good, and the data of very high quality. Being able to collect so many cruises with HR-ToF-AMS data is a really valuable contribution to the field. The paper is very suitable for ACP, but unfortunately major (big major) revision are needed:

Reply:

We thanks very much for referee's positive and constructive comments. The point-to-point responses are shown below:

- Introduction. Decide if you want to focus on the study area, or on the techniques, decide one flow and report it. At the moment there is confusion.

Reply:

Yes, the introduction part was confusing. We have rewritten the introduction to focus on the motivation of this study (referring to mainly the study area rather than the techniques). Also, we reorganized the abstract according to both referees' comments, in order to emphasize the key findings of this paper. According to referees' comments and co-authors' suggestion, we changed the title of this paper to "Organic aerosols over the Atlantic Ocean from 53°N to 53°S: similar contributions from ocean and long-range transport".

- There are 144 references, really there is no need to add all these references, suggestion to cut to 60 max.

Reply:

Thanks a lot! Indeed the amount of references is big. Considering that the other referee suggested to refer to more previous studies, we added some comparisons with historical

studies and removed the less useful references. Finally we shrank the references amount from 144 to 87 now. The existing references are considered to be necessary for better explaining the key findings of this paper.

- Figure S8. Factors F1-F4 and F1-F6 in PMF analysis need to be better described and named accordingly to the names of Factor 5 solution. Report also correlations among factors so the reader can understand how the factors evolve.

Reply:

To keep the focus of the manuscript, we added the description of factor evolution in the supplementary. In Figure S8, we put annotations for each factor to note the similarity between factors from 4- and 6-factor solutions and the selected solution, for example, the factor similar to MOOA was named as MOOA-like factor in 4- or 6- solutions. An additional plot following Figure S8 showed correlations among factors (Figure S9) to explain the reason of naming and the changing of the factors.

- The paper is very descriptive, and many papers are cited and referenced. There is no need. For example the whole section of Page 13 can be cut

Reply:

The authors went through the whole manuscript and tried best to remove the sentences which were descriptive and less useful for the key findings. In Page 13 (of original version), we reorganized the manuscript and removed most of the description of both MOA (now MOOA) and NOA (now MNOA) factors and shortened the discussion on them.

- pg 14 delete all topic of aminoacid, it creates confusion. These markers used are not unique of aminoacids.

Reply:

We agree with the comment and this is done.

- naming. perhaps you want to simplify the naming, for example the aPOA may simply

be anthropogenic organic aerosol (surely there will be a component that is secondary) and perhaps clearly stat that MOA POA and NOA are marine. NOA is marine, produced via secondary productions. Maybe start with "marine" or "anthopogenic" then "primary" or "secondary" then if it is Organic, nitrogen, MSA containing. Just a suggestion.

Reply:

We rethought about the naming of the OA factors and revised them for better indicating marine or anthropogenic sources. The MOA is now marine oxygenated OA (MOOA), NOA is changed to marine nitrogen-containing OA (MNOA), POA becomes marine hydrocarbon-like OA (MHOA), OOA is anthropogenic oxygenated OA (Anthr-OOA), and aPOA becomes combustion oxygenated OA (Comb-OOA).

- Overall it is advised that the senior scientists co-authoring this paper suggest how to improve the flow of the current manuscript.

Reply:

Thanks a lot for the suggestion. The senior scientists in the co-author list have read the manuscript and gave advices and suggestions on how to revise the paper. As you may see we reorganized the whole manuscript and rewrote many paragraphs in order to make the paper more clear and logical. Hope the efforts made the manuscript better.

I congratulate to the authors (both corresponding authors in particular) for the impressive dataset collected - once the flow of this paper is improved, it will make a very important contribution in the field.

Reply: Thank you very much for your encouraging comments!

[revised manuscript text omitted]
 (0.124,the NOA--). TheNOAclear peak in the, reaching thewhile the global radiation starts decreasingindicatingNOA factor is certainly composed oforganic products.~~ factor in New York City showing a diurnal pattern with maximum at noon (Sun et al., 2011). Both of these findings may indicate that secondary formation could be one of the possible pathways for MNOA generation.

N-containing OA (NOA) factors from PMF analysis have been found in many studies and can be related to various origins highly dependent on local  sources, such as Gentoo penguin hatching activities  (Schmale et al., 2013) and local primary (industrial) emissions (Aiken et al., 2009). During the R/V Polarstern campaign, the MNOA is correlated with neither eBC (R$^2$ = 0.17) nor NO$_3$ (R$^2$ = 0.06), excluding the possibility of combustion and anthropogenic (continental) sources. Meanwhile, high similarity (R$^2$ = 0.70) of the mass spectral profile is found between the R/V Polarstern MNOA and the NOA in Sun et al. (2011), who attributed that factor in New York City to marine and local industry emissions and stressed the possibility of gas-to-particle conversion via the reactions of acidic gases and the gaseous amines. It is worth noting that the characteristic C$_x$H$_y$N$^+$ fragments are actually different in these two studies: C$_3$H$_8$N$^+$ (m/z 58) and C$_2$H$_4$N$^+$ (m/z 42) dominated the C$_x$H$_y$N$^+$ 
[revised manuscript text omitted]
 3) an MHOA factor from primary marine emissions (19%), and two nonmarine factors, 1) an Anthr-OOA factor from continental outflow (19%) and 2) a Comb-OOA factor attributed to aged aerosol particles from combustion emissions mainly from biomass burning in Africa and maritime traffic over the Atlantic Ocean (30%). The MOOA factor shows prominent seasonality, with a higher contribution to the total OA mass in spring than in autumn; moreover, it shows a higher

20 contribution over the South Atlantic than the North  Atlantic in spring. This seasonality is, however, not observed for the other two marine factors. The MNOA and MHOA

25  both played a significant role in the clean regions with low particle mass concentration (e.g., in CR3 when the ship started from Punta Arenas). Continental influences on Atlantic aerosols were latitude-dependent during the R/V Polarstern measurements, represented by the Anthr-OOA and Comb-OOA factors. Both factors had dominant, even overwhelming, mass fractions when close to  land, e.g., Europe and West and Central

30 Africa , especially between 15°N and 15°S.

~~its mass concentration. The prominence of organics mass concentration has been found when the ship track was near the Europe or Africa. This is mainly linked to the continental outflow including anthropogenic pollutants and biomass burning emissions which can bring abundant organic aerosols. Moreover, the maritime traffic density is higher when closer to the continents, so the ship emissions would also contribute to MBL organic aerosols. Despite the marked continental influence on~~

5  During the R/V Polarstern cruises, marine sources contributed 51% to the total OA mass concentration, close to that of the nonmarine emissions (49%), reflecting the fact that continental emissions/human activities have a large influence on aerosols in the MBL 
[revised manuscript text omitted]

10 salt mass concentration is calculated according to the method from Bates et al. (2001) with the
11 equation: sea salt ($\mu$g m$^{-3}$) = Cl$^-$ ($\mu$g m$^{-3}$) + Na$^+$ ($\mu$g m$^{-3}$) $\times$ 1.47, the factor of 1.47 is the
12 seawater ratio of (Na$^+$ +K$^+$ +Mg$^{2+}$ + Ca$^{2+}$ +SO$_4^{2-}$ + HCO$_3^-$)/Na$^+$.

[Figure]

[Figure]

**Figure S7** Diagnostic plots: (a) Q/Q_exp ratio vs. number of factors, (b) Q/Q_exp vs. fPeak between

-1 and 1 in step of 0.2 for 5-factor solution, (c) Pearson's correlation coefficient R for time

series and mass spectra among 5 factors, (d) Q/Q_exp vs. seeds between 0 to 50 in step of 2, (e)

variation of mass fraction of each factor as a function of fPeak, (f) variation of mass fraction of each factor as a function of seeds, (g) comparison of total measured mass and reconstructed mass, (h) sum of the residuals of the fit, (i) $Q/Q_{exp}$ in time series, (j) $Q/Q_{exp}$ for each m/z, and (k) scaled residuals for each m/z, with horizontal bars for median, boxes for interquartile and sticks for 95% and 5% of points.

[Figure]

[Figure]

**Figure S8** Time series and mass spectra for OA components of (a, b) 4-factor solution, (c, d) 6-factor solution. (More explanation see Figure S9)

[Figure]

[Figure]

Figure S9 Correlation coefficients of time series and mass spectra for OA components of 4-factor solution and 6-factor solution comparing to 5-factor (selected) solution. The correlations among factors on time-series are shown in bars, while regarding mass spectra shown in cross with the same color code. Correlation coefficients reveal the evolution of the factors when the factor number changes: 4-factor solution already has stable MHOA-like, Comb-OOA-like, Anthr-OOA-like and MOOA-like factors, but MNOA factor is not identified; 6-factor solution keeps stable MHOA-like, Comb-OOA-like, MNOA-like and MOOA-like factors, while Anthr-OOA factor is separated into two parts. Note that the mass spectra of all factors show general resemblance due to high $CO^+$ and $CO_2^+$ ions. This results in higher $R^2$ on mass spectra than on time-series.

[Figure]

4 **Figure S**10 Similar variation between water temperature and MNOA (also colored in
5 water temperature) during 4 cruises.

[Figure]

[Figure]

2 **Figure S11** Latitude distribution of MHOA mass concentration (left axis), comparing to the

3 sea salt particle mass concentration (right axis). The MHOA is colored by wind speed (true

4 wind speed). Red boxes mark the cases in which the MHOA and estimated sea salt show similar

5 variation. Grey background indicates continental air masses, white one for marine air masses.

[Figure]

2 **Figure S12** Density map of the maritime traffic with Polarstern cruise tracks (black lines). The

3 background snapshot was taken from https://www.marinetraffic.com/en/ on May 2014 and

4 assumed to be similar to the situation in 2011 and 2012.

[Figure]

[Figure]

**Figure S13** Fire maps obtained from an online database of MODIS satellite (http://rapidfire.sci.gsfc.nasa.gov/firemaps/) , colored by Comb-OOA mass concentration during Polarstern cruises. The black arrows show the ship direction.

[Figure]
* * *

---

## Author Response (AR3)

**Response to Referees' Comments:**

Submitted on 03 Nov 2018

Anonymous Referee #1

Suggestions for revision or reasons for rejection (will be published if the paper is accepted for final publication)

Overall, with the latest revision, the paper is well formulated with the authors' valuable datasets and significant findings. I only have a few minor suggestions/questions for their consideration.

(1) Since the majority, also a novel part, of the paper is the source apportionment of OA, I would suggest the authors add "source apportionment" back to the title. This is also good for the paper to be accurately found via search engines.

Reply: Thank you very much for the suggestion. We changed the paper title to "Source Apportionment of the Organic Aerosol over the Atlantic Ocean from 53°N to 53°S: Significant Contributions from Marine Emissions and Long-Range Transport".

(2) Page 11, Line 1 - "a case study". Could the authors provide R2 of MOOA with MSA during the case period, and compare it with the overall R2 (0.83)? In addition, are you aware of any elevated marine bioactivities near the cruise, such as phytoplankton bloom?

Reply: Thanks. We added the required coefficient and compared it with the overall $R^2$ as shown in Page 11, Line 3 now: "The MOOA had strong correlation with MSA with $R^2$ of 0.81, almost the same as the overall coefficient ($R^2$ = 0.83). Note that the latter one is slightly higher mainly due to greater variation of both MOOA and MSA concentrations during all four cruises. This covariation also resulted in a quite stable MSA/MOOA ratio of 52% ± 9% during the MOOA dominating period."

During the case period (Nov. 18- Nov. 21, 2012), there was no visible phytoplankton bloom seen by eyes (at least no picture records). Unfortunately, the spatial distribution of the chlorophyll $a$ (Chl-$a$) mass concentration in this period is not full enough to indicate whether elevated marine biological activities occurred in this region, because the satellite observations were disturbed by clouds (as shown in Figure 1). However, it is worth noting that the marine secondary product tracer MSA showed maximum in the case period, which was repeatable in the same region and the same season (CR2 and CR4, Figure 2). This fact may suggest that there was seasonal and remarkable DMS-oxidation/marine SOA formation occurring in the selected region, even not indicated by the Chl-*a* concentration.

[Figure]

**Figure 1** Monthly (November, 2012, 2×2 km$^2$) average mass concentration of Chl-*a* (from MODIS aboard the satellite Aqua) with air mass origins (grey lines, 5-day back trajectories at 950hPa), the ship track is shown in orange line and red box provides the range of selected MOOA dominating period.

[Figure]

**Figure 2** Spatial distribution of the MSA mass concentration in the four cruises (Huang et al., 2017). Background plots provide simultaneous Chl-*a* mass concentration (rolling 32-day average), obtained from MODIS aboard the satellite Aqua. Air mass back trajectories are for the past 5 days at 950 hPa. The box with red dots indicates the selected MOOA dominating period in CR4 and the same region in CR2.

(3) Page 11, Line 14 - "...and most MOOA peaks were associated with marine air masses (Figure 3)". The description might be true, but it is actually not obvious as in Figure 3. Can you conduct a statistical test to examine it? although one simply way is to compare the average MOOA from marine vs. continental air masses.

Reply: Thanks for your comments and suggestion. We calculated the average concentration of MOOA from each air mass group, and added one sentence to quantify the difference, as shown in Page 11, Line 17 now: "
[revised manuscript text omitted]

salt mass concentration is calculated according to the method from Bates et al. (2001) with the
equation: sea salt ($\mu$g m$^{-3}$) = Cl$^-$ ($\mu$g m$^{-3}$) + Na$^+$ ($\mu$g m$^{-3}$) $\times$ 1.47, the factor of 1.47 is the
seawater ratio of (Na$^+$ +K$^+$ +Mg$^{2+}$ + Ca$^{2+}$ +SO$_4$$^{2-}$ + HCO$_3$$^-$)/Na$^+$.

[Figure]

[Figure]

**Figure S7** Diagnostic plots: (a) Q/Q$_{exp}$ ratio vs. number of factors, (b) Q/Q$_{exp}$ vs. fPeak between

-1 and 1 in step of 0.2 for 5-factor solution, (c) Pearson's correlation coefficient R for time series and mass spectra among 5 factors, (d) Q/Q$_{exp}$ vs. seeds between 0 to 50 in step of 2, (e)

variation of mass fraction of each factor as a function of fPeak, (f) variation of mass fraction of each factor as a function of seeds, (g) comparison of total measured mass and reconstructed mass, (h) sum of the residuals of the fit, (i) Q/Q$_{exp}$ in time series, (j) Q/Q$_{exp}$ for each m/z, and (k) scaled residuals for each m/z, with horizontal bars for median, boxes for interquartile and sticks for 95% and 5% of points.

[Figure]

[Figure]

**Figure S8** Time series and mass spectra for OA components of (a, b) 4-factor solution, (c, d)
6-factor solution. (More explanation see Figure S9)

[Figure]

[Figure]

**Figure S9** Correlation coefficients of time series and mass spectra for OA components of 4-factor solution and 6-factor solution comparing to 5-factor (selected) solution. The correlations among factors on time-series are shown in bars, while regarding mass spectra shown in cross with the same color code. Correlation coefficients reveal the evolution of the factors when the factor number changes: 4-factor solution already has stable MHOA-like, Comb-OOA-like, Anth-OOA-like and MOOA-like factors, but MNOA factor is not identified; 6-factor solution keeps stable MHOA-like, Comb-OOA-like, MNOA-like and MOOA-like factors, while Anth-OOA factor is separated into two parts. Note that the mass spectra of all factors show general resemblance due to high $CO^+$ and $CO_2^+$ ions. This results in higher $R^2$ on mass spectra than on time-series.

[Figure]

**Figure S10** Similar variation between water temperature and MNOA (also colored in water
temperature) during four cruises.

[Figure]

[Figure]

**Figure S11** Latitude distribution of MHOA mass concentration (left axis), comparing to the sea salt particle mass concentration (right axis). The MHOA is colored by wind speed (true wind speed). Red boxes mark the cases in which the MHOA and estimated sea salt show similar variation. Grey background indicates continental air masses, white one for marine air masses.

[Figure]

**Figure S12** Density map of the maritime traffic with Polarstern cruise tracks (black lines). The
background snapshot was taken from https://www.marinetraffic.com/en/ on May 2014 and
assumed to be similar to the situation in 2011 and 2012.

[Figure]

**Figure S13** Fire maps obtained from an online database of MODIS satellite
(http://rapidfire.sci.gsfc.nasa.gov/firemaps/) , colored by Comb-OOA mass concentration
during Polarstern cruises. The black arrows show the ship direction.